# Hazard Assessment Comparison of Tazhiping Landslide Before and After Treatment using the Finite Volume Method

Dong Huang [1], YuanJun Jiang[1]*, JianPing Qiao[1], Meng Wang[1]

1. Key Laboratory of Mountain hazards and Surface process, Institute of Mountain hazards and Environment, Chinese Academy of Science, Chengdu 610041, China
*Corresponding author ( yuanjun.jiang.civil@gmail.com).

**Abstract:** Through investigation and analysis of geological conditions and mechanical parameters of the Taziping landslide, Finite Volume Method coupling with Vollmy model is used to simulate the landslide mass movement process. The present paper adopts the numerical approach of RAMMS and the GIS platform to simulate the mass movement process before and after engineering treatment. This paper also provides the conditions and characteristic variables of flow-type landslide in terms of flow height, velocity, and stresses. The 3D division of hazard zones before and after engineering treatment was also mapped. The results indicate that the scope of hazard zones decreased after engineering treatment of the landslide. Compared with the case of before engineering treatment, the extent of high-hazard zones was reduced by about 2/3, and the characteristic variables of the mass movement in the case of after treatment decreased to 1/3 of those in the case of before treatment. Despite having engineering treatment, the Taziping landslide still poses significant potential threat to the nearby residences. Therefore, it suggests that the houses located in high-hazard zones should be relocated or reinforced for protection.

**Keywords**: finite volume method; rheological model; motion feature parameters; hazard assessment

## 1. Introduction

The hazards of a landslide include scope of influence (i.e., source area, possible path area, and backward and lateral expansion area) and secondary disasters (i.e., reservoir surge, blast, and landslide-induced barrier lake). A typical landslide hazard assessment aims to propose a systematic hazard assessment method with regard to a given position or a potential landslide. Current research on typical landslide hazard assessment remains immature, and there are multiple methods for interpreting landslide hazards. To be specific, the scope of influence prediction of a landslide refers to deformation and instability characteristics such as sliding distance, movement speed, and bulking thickness range. The movement behavior of a landslide mass is related to its occurrence, sliding mechanisms, mass characteristics, sliding path, and many other factors. Current landslide movement prediction methods include empirical prediction and numerical simulation.

**Empirical prediction method:** The empirical prediction method involves

analyzing landslide flow through the collection of landslide parameters in the field. It further consists of the geomorphologic method (Costa, 1984; Jackson et al., 1987; Scott et al., 1993), the geometric change method (Finlay et al., 1999; Michael-Leiba et al., 2003), and the volume change method (Fannin et al., 2001). Empirical models are commonly simple and easy to apply, and the required data are easy to obtain as well. **Numerical simulation method:** Numerical simulation methods are further divided into the continuous deformation analysis method (Hungr, 1995; Evans et al., 2009; Wang, et al., 2016), the discontinuous deformation analysis method (Shi, 1988), and the simplified analytical simulation method (Christen et al., 2010a; Sassa, 2010; Bartelt et al., 2012; Du et al., 2015). The numerical simulation method expresses continuous physical variables using the original spatial and temporal coordinates with geometric values of discrete points. Numerical simulations follow certain rules to establish an algebraic equation set in order to obtain approximate solutions for physical variables.

Empirical prediction models only provide a simple prediction of the sliding path. Due to the differences in geological environments, empirical prediction models commonly have low generality. Landslides move downslope in many different ways (Varnes, 1978). In addition, landslides can evolve into rapidly travelling flows, which exhibit characteristics of debris flows on unchannelized or only weakly channelized hillslopes. The geomorphic heterogeneity of rapid shallow landslides, such as hillslope debris flows, is larger than observed in channelized debris flows; however many of these flows can be successfully modelled using the Voellmy-fluid friction (Christen et al., 2012). The selection of model parameters remains one of the fundamental challenges for numerical calculations of natural hazards.

The continuous deformation method has the advantage of an extremely strong replication capability, but it is not recommended when analyzing flow-type landslides, lahars, or debris flows because of complicated rheological behaviors (Iverson et al., 1997, 2001; Hungr et al., 2001; Glade 2005; Portilla et al., 2010; Chen et al., 2014). The fluid mechanics-based discontinuous deformation method has several shortcomings such as, great computational burden, difficult parameter selection, and difficult 3D implementation. The simplified analytical simulation method fully takes into account the flow state properties of landslides before introducing a rheological model and can easily realize 3D implementation on the GIS platform. On that account, this paper adopted the continuous fluid mechanics-based finite volume method (simplified analytical simulation method). We introduce a rheological model on the basis of using mass as well as momentum and energy conservation to describe the movement of landslides. We also employed GIS analysis to simulate the entire movement process of Taziping landslide and map the 2D division of hazard zones.

## 2. Methods

### 2.1 Kinetic analysis method

Adopting the continuous fluid mechanics-based finite volume method, this paper took into account erosion action on the lower surface of the sliding mass and the

change in frictional resistance within the landslide-debris flow in order to establish a
computational model. The basic idea is to divide the calculation area into a series of
non-repetitive control volumes, ensuring that there is a control volume around each
grid point. Each control volume is then integrated by the unresolved differential
equation in order to obtain a set of discrete equations. The unknown variable is the
numerical value of the dependent variable at each grid point. To solve the integral of a
control volume, we make a hypothesis about the change rule of values among grid
points, that is, about their piecewise distribution profile. The finite volume method
can satisfactorily overcome the finite element method's weakness of slow calculation,
and solve the problem of complex region processing. Thus, we adopted the finite
volume method to establish the kinematic model for the landslide flow process.
The core of the finite volume method is domain discretization. The finite volume
method uses discrete points as a substitute for continuous space. The physical
meaning of the discrete equation is the conservation of the dependent variable in a
finite control volume. Establishment of the conservation equation is based on the
continuous movement model, that is, the continuity hypothesis about landslide
substances. We divided the landslide mass into a series of units and made the
hypothesis that each unit has consistent kinematic parameters (speed at a depth,
density, etc.) and physical parameters (Fig.1). We also established an Eulerian
coordinate system-based conservation equation with regard to each control volume.

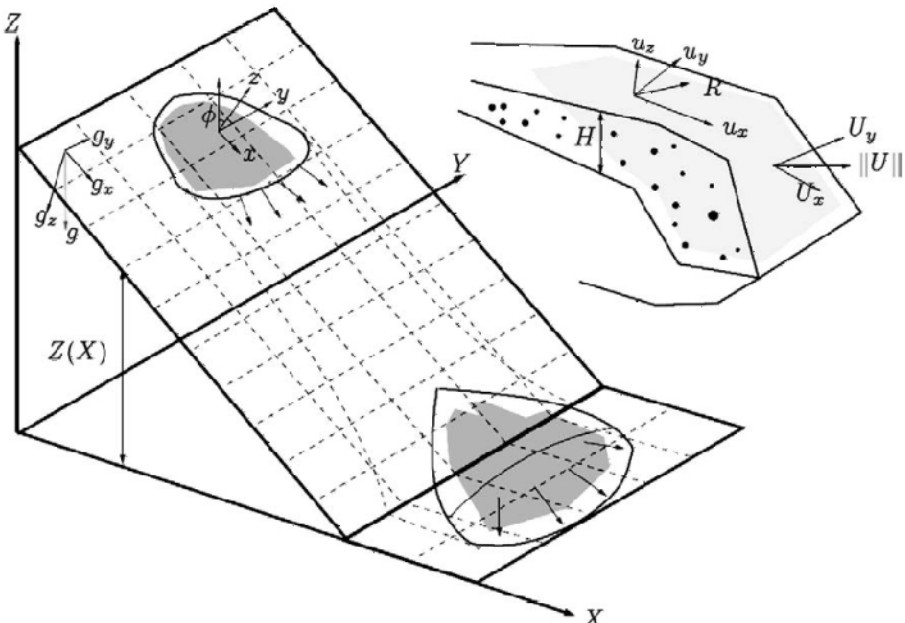


Fig.1 Schematic diagram of finite volume discretization (Christen et al., 2010a).
**2.2 Control equation**
The computational domain is defined as directions $x$ and $y$, and the
topographic elevation is given the coordinate $z(x,y)$. $H(x,y,t)$ is assumed as the
change relationship of landslide thickness with time; $U_x(x,y,t)$ and $U_y(x,y,t)$
respectively represent the mean movement speeds along directions $x$ and $y$ at
moment t; $n_x = U_x / \sqrt{U_x^2 + U_y^2}$ and $n_y = U_y / \sqrt{U_x^2 + U_y^2}$ represent the cosinoidal and
sinusoidal flow vectors of the landslide on the plane $x$-$y$. The mean flow speed of
substances is defined as $U = \sqrt{U_x^2 + U_y^2}$ .
Thus, the mass balance equation becomes:
$$\partial_t H + \partial_x (HU_x) + \partial_y (HU_y) = \dot{Q} \tag{1}$$
wherein, $\dot{Q}(x, y, t)$ represents the change rate (entrainment rate) of landslide
volume with time.
Assuming that $l(x, y, t)$ represents the movement distance of the landslide with
time, we can obtain:
$$\dot{Q} = \begin{cases} 0 & if & h_i = 0 \\ \dfrac{\rho_i}{\rho_a} h_i \dfrac{U}{l} & if & k_i l \geq h_i \\ \dfrac{\rho_i}{\rho_a} k_i U & if & k_i l < h_i \end{cases} \tag{2}$$

wherein, $h_i$ represents the thickness of the $i$th layer of the landslide in the
movement process; $\rho_i$ represents the density of the $i$th layer of the landslide in the
movement process; $\rho_a$ represents the density of the landslide; the dimensionless
parameter $k_i$ represents the entrainment rate.
The momentum balance equation is:
$$\partial_t (HU_x) + \partial_x (HU_x^2 + \frac{g_z k_{a/p} H^2}{2}) + \partial_y (HU_x U_y) = S_{gy} - S_f(R)[n_x] \tag{3}$$
$$\partial_t (HU_y) + \partial_y (HU_y^2 + \frac{g_z k_{a/p} H^2}{2}) + \partial_x (HU_x U_y) = S_{gx} - S_f(R)[n_y] \tag{4}$$
wherein, $S_{gx} = g_x H$ and $S_{gy} = g_y H$ represent the dynamic components of the
acceleration of gravity in directions $x$ and $y$; $g = (g_x \quad g_y \quad g_z)$ represents the
vector of the acceleration of gravity; $k_{a/p}$ represents the pressure coefficient of soil;
$\rho_a$ represents the density of the landslide; the dimensionless parameter $k_i$
represents the entrainment rate; $S_f(R)$ represents the frictional resistance.

136       The kinetic energy balance equation is:

$$\partial_t(HR) + \partial_x(HRU_x) + \partial_y(HRU_y) = \dot{P} - \dot{D} \tag{5}$$

wherein, $R(x,y,t)$ represents the random mean kinetic energy of the landslide;

$\dot{P}(x,y,t)$ and $\dot{D}(x,y,t)$ represent the random increased kinetic energy and decreased
kinetic energy of the landslide.
**2.3 Constitutive relationship**

The improved Voellmy rheological model is applied in the computational

simulation of the landslide. See the computational formula below:
$$S_f = \frac{u_i}{\|U\|}\left(h\mu g_z + R_t U^2 + R_\zeta U^2\right) \tag{6}$$

$$R_t = \mu h \frac{U^T K U}{U^2}, R_\zeta = \frac{g}{\zeta} \tag{7}$$

wherein, $u_i/\|U\|$ represents the unit vector in the movement direction of the

landslide; $\mu$ represents the Coulomb friction coefficient, and is related to $R(x,y,t)$,
the random mean kinetic energy of the landslide; $R_t$ represents the gravity-related
frictional force coefficient; $K$ represents the substrate surface curvature; $\zeta$
represents the viscous friction coefficient of the "turbulent flow".
**2.4 HLLE-Heun numerical solution**

Synthesizing control equations (1), (3), (4) and (5), we can obtain the simplified

form of the nonlinear hyperbola equation:
$$\partial_t V + \nabla \cdot F(V) = G(V) \tag{8}$$

$$V = \begin{pmatrix} H \\ HU_x \\ HU_y \\ HR \end{pmatrix} \qquad G(V) := \begin{pmatrix} \dot{Q} \\ S_{gx} - S_{fx} \\ S_{gy} - S_{fy} \\ \dot{P} - \dot{D} \end{pmatrix}$$


$$F(V) = \begin{pmatrix} HU_x & HU_y \\ HU_x^2 + g_z k_{a/p} \dfrac{H^2}{2} & HU_x U_y \\ HU_x U_y & HU_y^2 + g_z k_{a/p} \dfrac{H^2}{2} \\ HRU_x & HRU_y \end{pmatrix}$$

wherein, $V(x,y,t)$ represents a vector equation consisting of four unknown
vector variables; $F(V)$ represents the flux function; $G(V)$ represents the source
term. Based on the HLLE equation of the finite volume method and the quadrilateral
grid, the node layout can adopt the grid center pattern, and the normal flux along one
side of the control volume can be represented by the flux at the center of the side. The
finite volume discretization adopting the control volume as unit is depicted in Fig.1;
the Gauss theorem can be followed for the integration of equation (8), wherein $C_i$
represents the unit volume; after converting the volume integral flux function $F(V)$
into the curved surface integral, we can obtain:

$$\int_{C_i} \partial_t V dx + \iint_{\partial C_i} F(V) \cdot n_i d\sigma = \int_{C_i} G(V) dx \qquad (9)$$

wherein, $n_i$ represents the outward normal direction vertical to unit $C_i$ at the
boundary; through adopting the HLL format for the discretization of surface integral,
the following simplified form can be obtained:

$$V_i^{(*)} = V_i^{(n)} + \frac{\Delta t}{A_{C_i}} \Delta F_i^{(HLL)}\left(V^{(n)}\right) \qquad (10)$$


$$V_i^{(**)} = V_i^{(*)} + \frac{\Delta t}{A_{C_i}} \Delta F_i^{(HLL)}\left(V^{(*)}\right) \qquad (11)$$


$$V_i^{(n+1)} = \frac{1}{2}\left(V_i^{(n)} + V_i^{(**)}\right) \qquad (12)$$

wherein, $V_i^{(n)}$ represents the mean value of unit variables at moment $t^{(n)}$; $V^{(n)}$
represents the mean value of the entire grid at moment $t^{(n)}$; $\Delta t := t^{(n-1)} - t^{(n)}$ represents
the calculated time step; $A_{C_i}$ represents the area of unit $C_i$; $\Delta F_i^{(HLL)}$ represents the
approximate value of the curved surface integral, as shown below:

$$\Delta F_i^{(HLL)}\left(V^{(n)}\right) := -\sum_{j=1}^{4} F_{ij}^{(HLL)}\left(V^{(n)}\right) n_{ij} \Delta X \qquad (13)$$

wherein, $n_{ij}$ represents the outward normal direction of the $i$ th unit at
boundary $j$; the flux calculation term $F_{ij}^{(HLL)}\left(V^{(n)}\right)$ represents the approximate
solution mode of the Riemann problem of the $i$th unit at boundary $j$; see the
computational formula below:
$$F_{ij}^{(HLL)}\left(V^{(n)}\right) = \begin{cases} F\left(V_L^{(n)}\right) & 0 \le S_L \\ \dfrac{S_R F\left(V_L^{(n)}\right) - S_L F\left(V_R^{(n)}\right) + S_R S_L F\left(V_R^{(n)} - V_L^{(n)}\right)}{S_R - S_L} & S_L \le 0 \le S_R \\ F\left(V_R^{(n)}\right) & S_R \le 0 \end{cases} \qquad (14)$$

wherein, $V_L^{(n)}$ and $V_R^{(n)}$ respectively represent the approximate values of $V^{(n)}$
on both sides of boundary $j$ of the $i$th unit; $S_L$ and $S_R$ respectively represent the
wave speeds on the left and right sides. Refer to the computational method described
by Toro (1992). In addition, the gradient magnitude in the original second-order
difference equation can be limited through multiplication with the flux limiter, and the
second-order format of the TVD property can be constructed to avoid the occurrence
of numerical oscillation. Refer to the specific method described by LeVeque (2002).
In this paper a numerical solver within RAMMS is used, which was specifically
designed to provide landslide (avalanche) engineers with a tool that can analyze
problems with two-dimensional depth-averaged mass and momentum equations on
three-dimensional terrain using both first and second-order finite volume methods
(Christen et al., 2010b).Therefore, the finite volume method is adopted to analyze the
the flow-type (high mobility, high velocity, large scope of risks, etc.) of the landslide
mass movement process. The present paper adopts the numerical approach of
RAMMS and the GIS platform to simulate the mass movement process before and
after treatment. The landslide depositional characteristics and the mass movement
conditions can be combined to provide a scientific basis for engineering prevention ,
control, and forecast risk assessments for these kinds of disasters.

## 3. Study area and data

### 3.1 Taziping landslide

The Taziping landslide is located southeast of the Hongse Village, Hongkou
Town, Dujiangyan City of Sichuan Province. The site is located at (E103°37′46″,
N31°6′29″), 68 km west Chengdu City and 20 km from the Dujiangyan Urban
District (Fig. 2). Its geomorphic unit is a middle-mountain tectonic erosional area on
the north bank of the Baisha River Valley. The Taziping Landslide is a large-scale
colluvial layer landslide triggered by the Wenchuan Earthquake (Fig. 3). It has a
gradient of 25°-40° with an average gradient of 32°. The landslide has an apparent
round-backed armchair contour with a steep rear edge, which has a gradient of
35°-50° and an elevation of about 1,370 m. The front edge is located on the south side
of the mountain road, and has an elevation of about 1,007 m. The landslide has an

elevation difference of about 363 m, and a main sliding direction of 124°NE. The landslide mass forms an irregular semi-elliptical shape, and has a length of about 530 m, an average width of 145 m and an area of approximately $7.68 \times 10^4$ m$^2$. The landslide mass is composed of gravelly soil and is covered on by silty clay mingled with gravel. In terms of spatial distribution, the landslide is thick in the middle and thin on the lateral edges, has a thickness of 20-25 m and a volume of approximately $1.16 \times 10^6$ m$^3$. During the earthquake, the landslide mass slid to cover the northern mountain slope of the Hongse Village Miaoba settlement. The landslide has an apparent front edge boundary, and there is also a swelling deformation (Fig. 4).

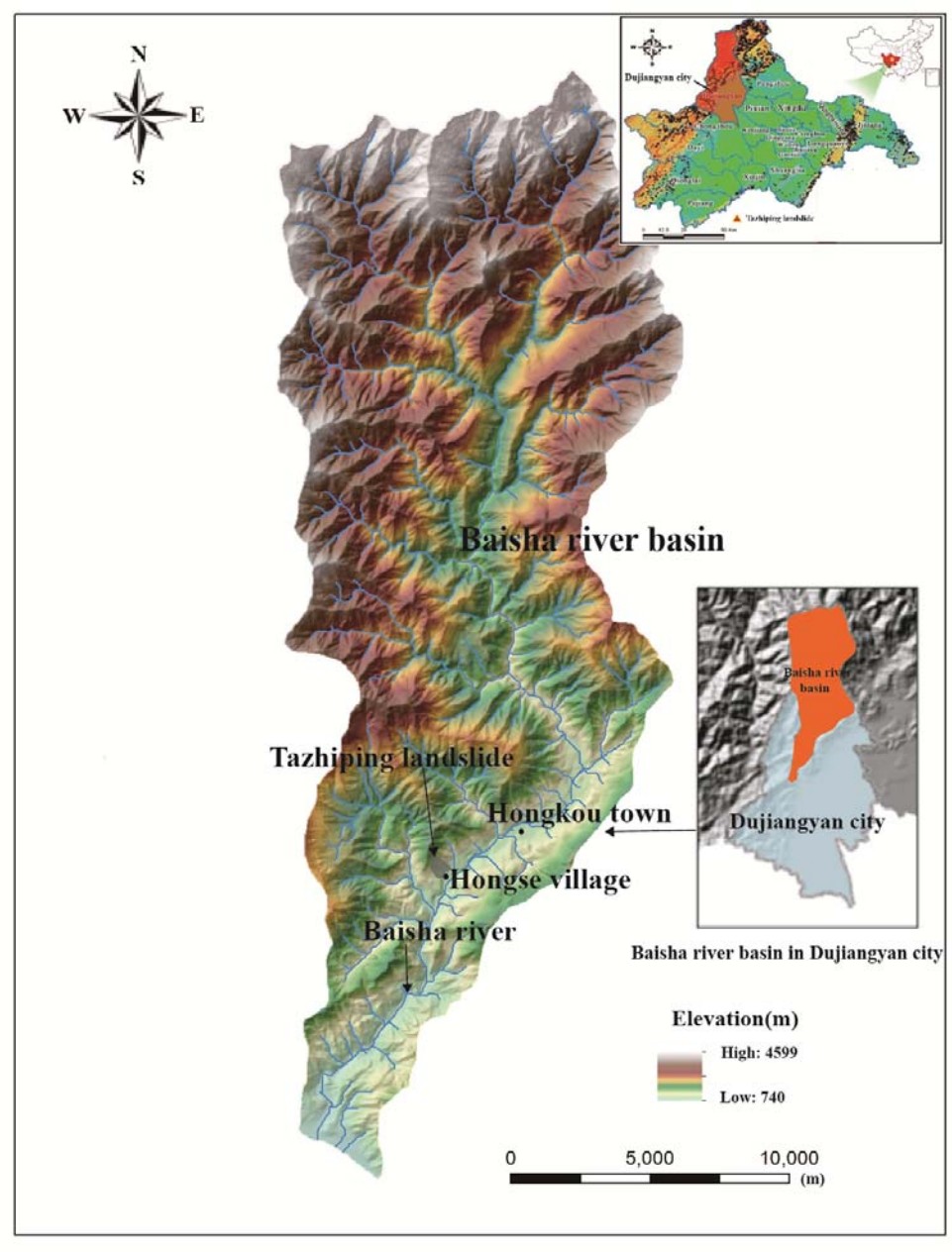

Fig.2 Location of Tazhiping landslide, Baisha river basin, Dujiangyan city (the landslide was triggered by Wenchuan Ms 8.0 earthquake on May 12, 2008)

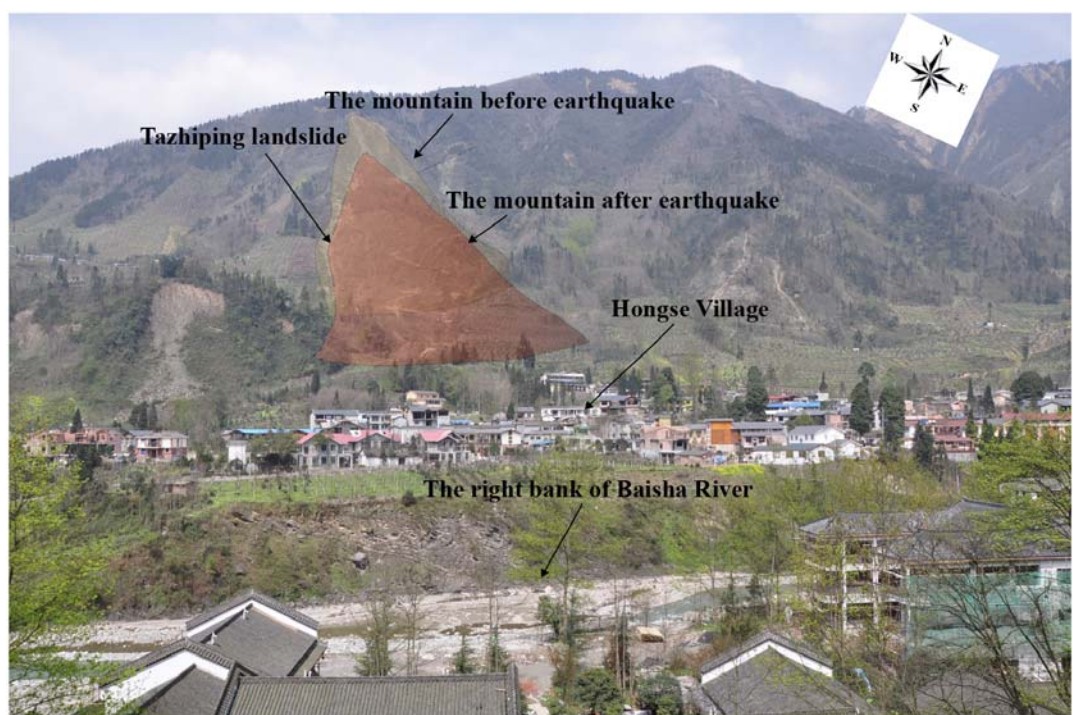


Fig.3 Taziping Landslide

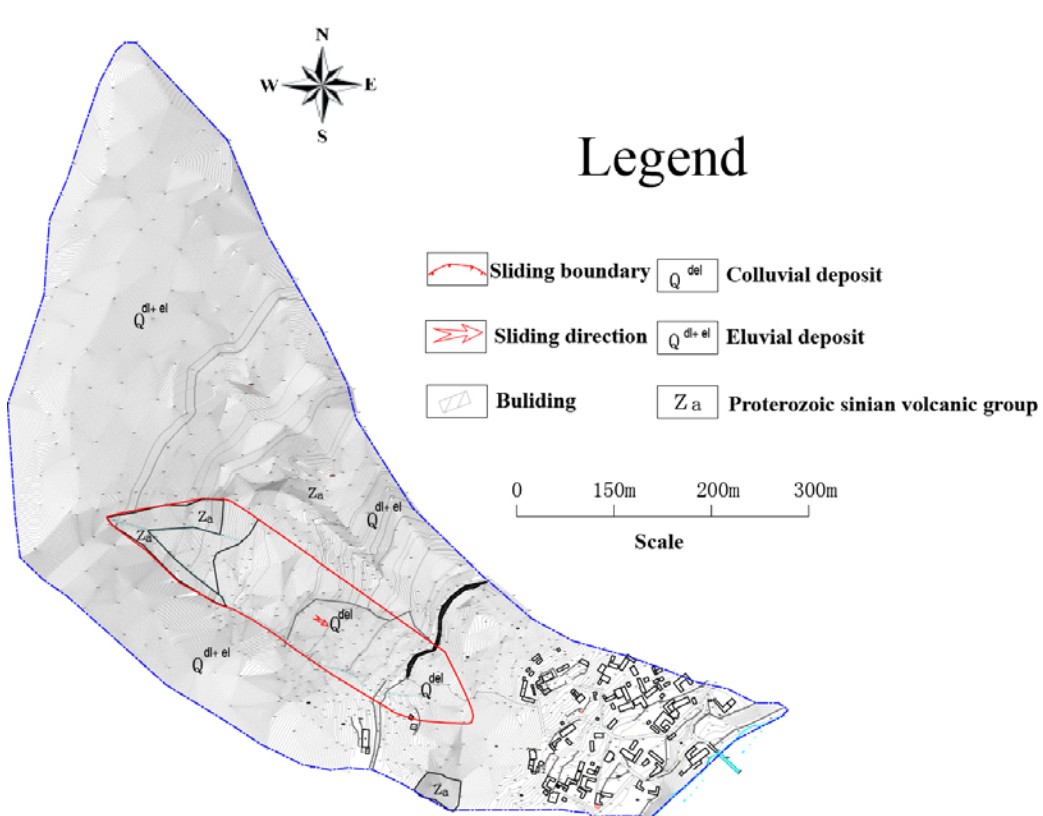


Fig.4 Plane sketch of the Tazhiping landslide

After the Wenchuan Earthquake, the massive colluvial deposits covered the
mountain slope. The colluvium is 0.5-5.0 m thick at the top of the slide and is
composed of rubble and gravel. The mass consists of a small amount of fine gravel,

which is composed of gray or grayish-green andesite with a clast of 20-150 cm. Field surveys indicate that the rubble in the surface layer has a maximum diameter exceeding 2 m, and that fine gravel is loosely intercalated with the rubble. A small amount of yellowish-brown and gray-brown silty clay mixed with 5-40% of non-uniformly distributed rubble composed the first 5-10 m of the slide. From 10-25 m deep, there is a wide distribution of gravelly soil. The soil is grayish-green or variegated in color, is slightly compact and non-uniform, and has a rock fragment content of about 50%. The parent rock of the rock fragments is andesite, filled with silty clay or silt (Fig.5). Table 1 shows the parameters of the surface gravelly soil of the landslide mass based on the field sampling.

Tab.1 Parameters of   surface soil of Taziping Landslide

| Internal friction angle (°) | | Cohesion (kPa) | Relative compactness | Natural void ratio | Dry density (kN·m⁻³) | Specific gravity (g·cm⁻³) |
|---|---|---|---|---|---|---|
| Peak | Residual | | | | | |
| 27.5 | 23 | 20.5 | 53% | 0.789 | 15.357 | 2.492 |

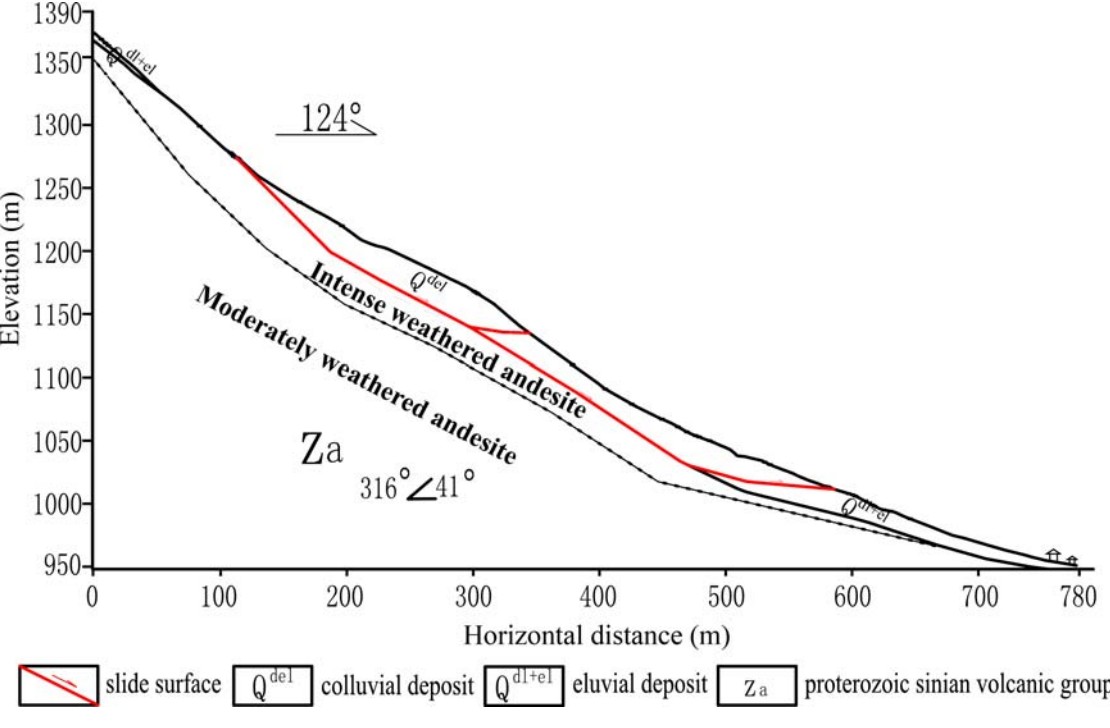

Fig.5 Geological profile of the Taziping Landslide

The landslide is an unconsolidated mass containing relatively large amounts of crushed stones and silty clay (Fig.6). Its loose structure and strong permeability facilitate infiltration of surface water. The Wenchuan earthquake aggravated the deformation of the landslide making deposits more unconsolidated, further reducing the stability of the landslide mass. During persistent rainfall, surface water infiltrates the landslide slope resulting in increased water pressure within the landslide mass and

reduced shear strength on the sliding surface. Thus, rainfall constitutes the primary
inducing factor of the upper Taziping landslide. After infiltrating the loose layer, water
saturates the slope increasing the dead weight of the sliding mass and reducing the
shear strength of soil in the sliding zone. Infiltration into the landslide mass also
increases the infiltration pressure of perched water, drives deformation, and poses a
great threat to villages located at the front of the landslide. Slide-resistant piles and
backfill were place at the toe of the slope in order to reduce the hazards of future
slides. The slide-resistant piles have enhanced the overall stability of the slope,
however, under heavy rainfall the upper unconsolidated landslide deposits may cut
out from the top of the slide-resistant piles.

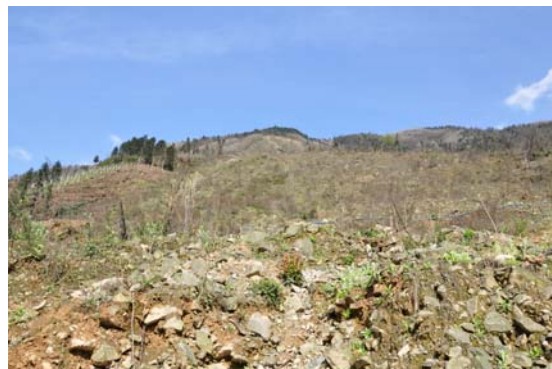 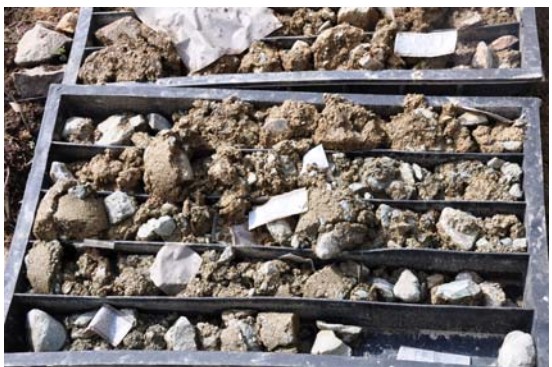

(a) Material on the landslide surface     (b) Material in the shear zone

Fig.6    Photographs showing colluvial deposit cover on the mountain slope
Therefore we simulate possible movement states of the Taziping landslide before
and after treatment with slide-resistant piles, comparatively analyzed the kinetic
parameters in the movement process, and mapped the 2D division of hazard zones.

**3.2 Hazard prediction before treatment**
It was assumed that the landslide was damaged before engineering treatment.
According to field investigation, the sliding mass had an estimated starting volume of
about 600,000m³ and a mean thickness of 8m. Based on the survey report and field
investigation (Hydrologic Engineering and Geological Survey Institute of Hebei
Province, 2010), we adopted the survey parameters of Tab.2 for the simulated
calculation. These parameters were obtained from laboratory or small-scale
experiments and back-analyses of relatively well-documented landslide cases. The
unit weigh $\gamma = 20.8kN \cdot m^{-3}$ is from small-scale conventional
triaxial test experiments in laboratory. In addition, we selected the coulomb friction
coefficient $\mu = 0.45$ and viscous friction coefficient $\zeta = 500m \cdot s^{-2}$ in accordance
with back-analyses of well-documented landslide cases (Cepeda et al., 2010; Du et al.,
2015). The erosional entrainment rate selected was the minimum value $k_i = 0.0001$
in the RAMMS program.

Tab.2 Model calculation parameters

| Unit weight $\gamma(kN \cdot m^{-3})$ | Coulomb friction coefficient $\mu$ | Viscous friction coefficient $\zeta(m \cdot s^{-2})$ | Erosional entrainment rate $k_i$ |
|---|---|---|---|
| 20.8 | 0.45 | 500 | 0.0001 |


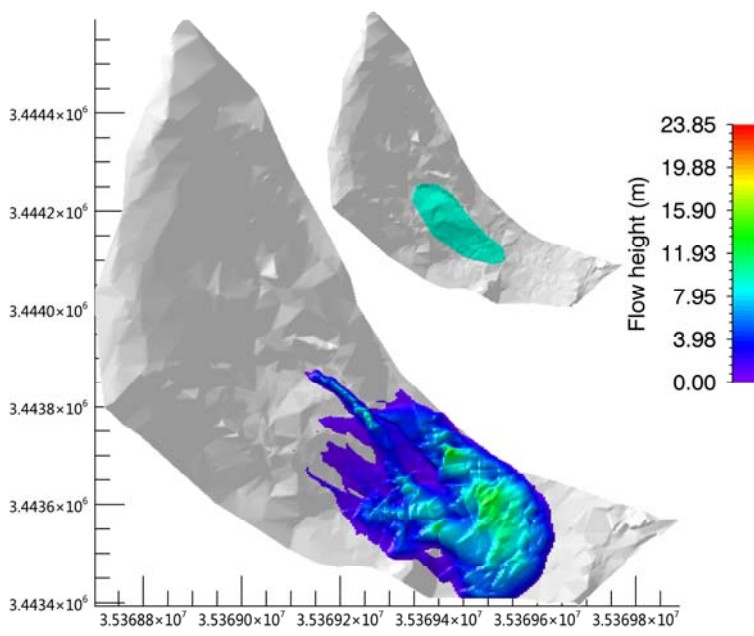

(a) Flow height

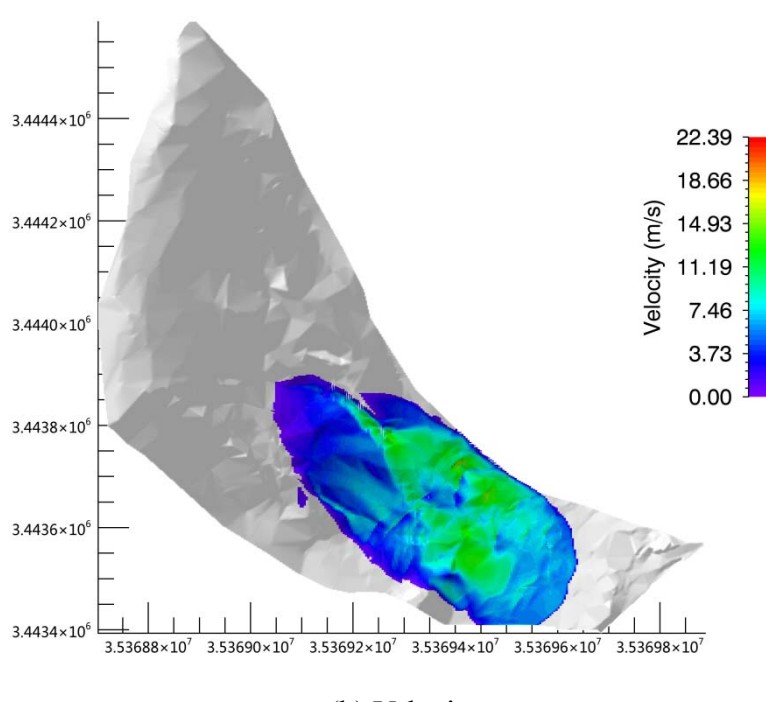


(b) Velocity

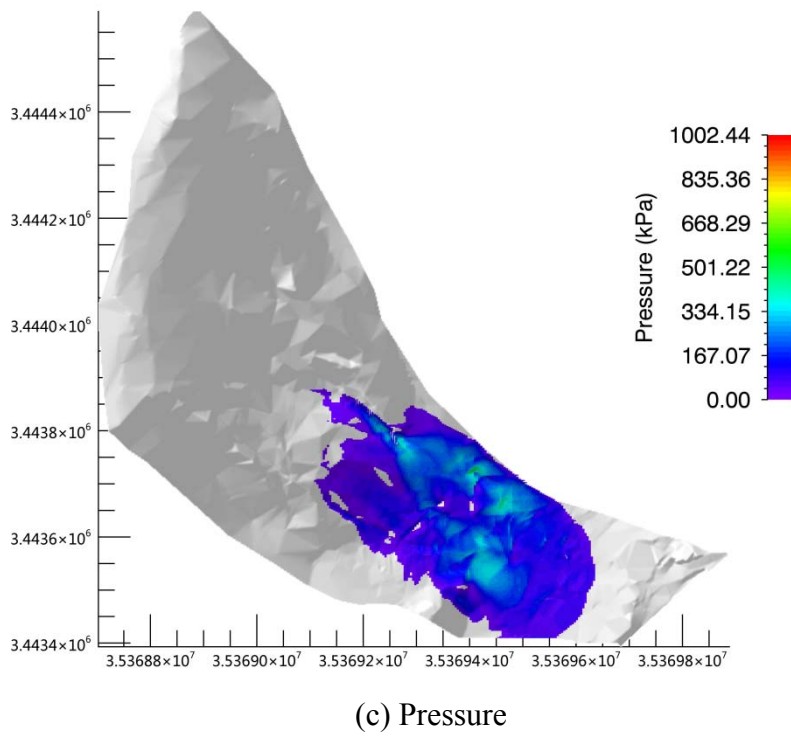

(c) Pressure

Fig. 7 Movement characteristic parameters of the Taziping landslide (before

treatment)

See the kinematic characteristic parameters of the landslide deposits in Fig.7. The colored bar shows the maximum values of the kinematic process for a given time step. As shown by the calculation results, deposits accumulated during the landslide movement process had a maximum flow height of 23.85m, located around the surface gully of the middle and upper slope. The middle and lower section of the landslide deposit had a flow height of about 5-10m; the middle and lower movement velocity of the landslide ranged from 3m/s and 7m/s; the landslide had a mean pressure of about 500kPa, and the pressure of the middle and lower deposits was about 200kPa. Thus, three-story and lower houses within the deposition range might be buried (The building is 3m high on each floor), and it was further suggested that the design strength of the gable walls of houses on the middle and upper parts of the deposit be increased above 300kPa.

**3.3 Hazard prediction after treatment**

After fully accounting for the slide-resistant piles and mounds, we introduced the Morgenstern-Price method (Morgenstern et al., 1965) to calculate the stability coefficient of Taziping landslide after treatment. The method was determined with an iterative approach by changing the position of the sliding surface until failure of the dumpsite (Fig.8). The physico-mechanical parameters under a saturated state (Hydrologic Engineering and Geological Survey Institute of Hebei Province, 2010) were adopted to search for the sliding plane of the landslide.

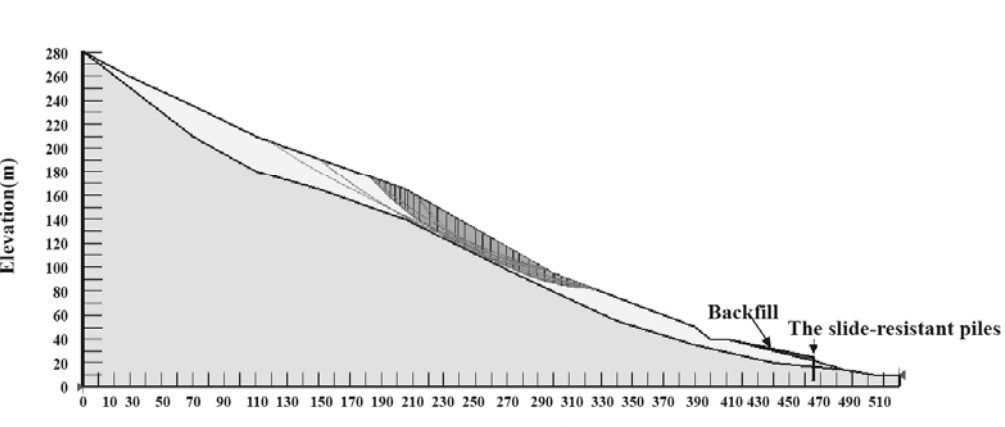

Fig.8 Search for the sliding plane of the Taziping landslide (before treatment)

Based on numerical analysis, the Taziping landslide stability coefficient is 0.998. Under rainfall conditions, the middle area of the Taziping landslide was unstable. Loose deposits in the middle part of the landslide might convert into a high-water landslide and cut out from the top of the slide-resistant piles. In the damaged area, the slope had a rear edge wall elevation of about 1,170m. Its front edge was located on the south side of the mountain road, with an elevation of 1,070-1,072m and a length of 182m. Thus, the scale of the rainfall-damaged is estimated to be about 250,000m$^3$, with a mean thickness of about 6m. The parameters in Tab.2 were again adopted for the simulated calculation.

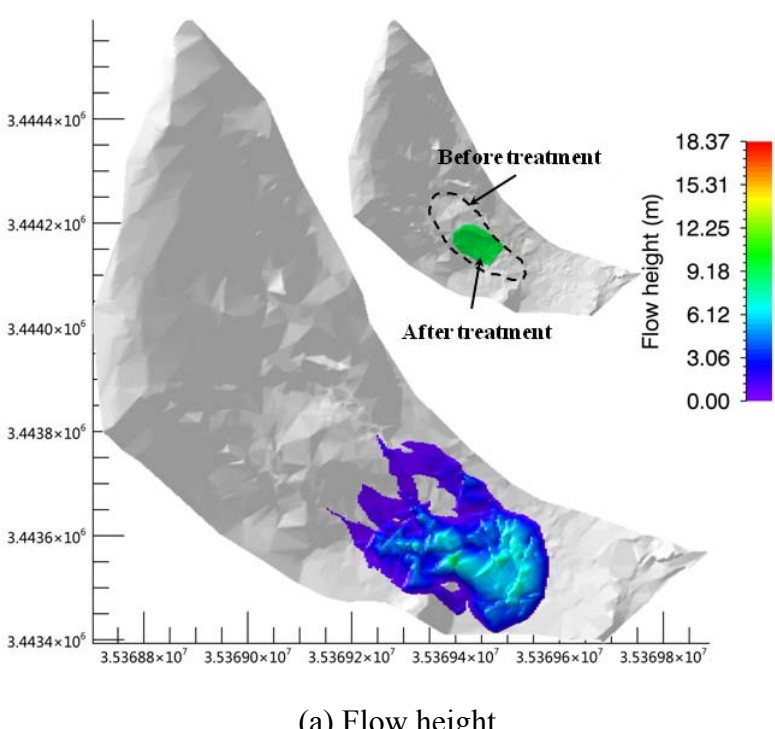


(a) Flow height

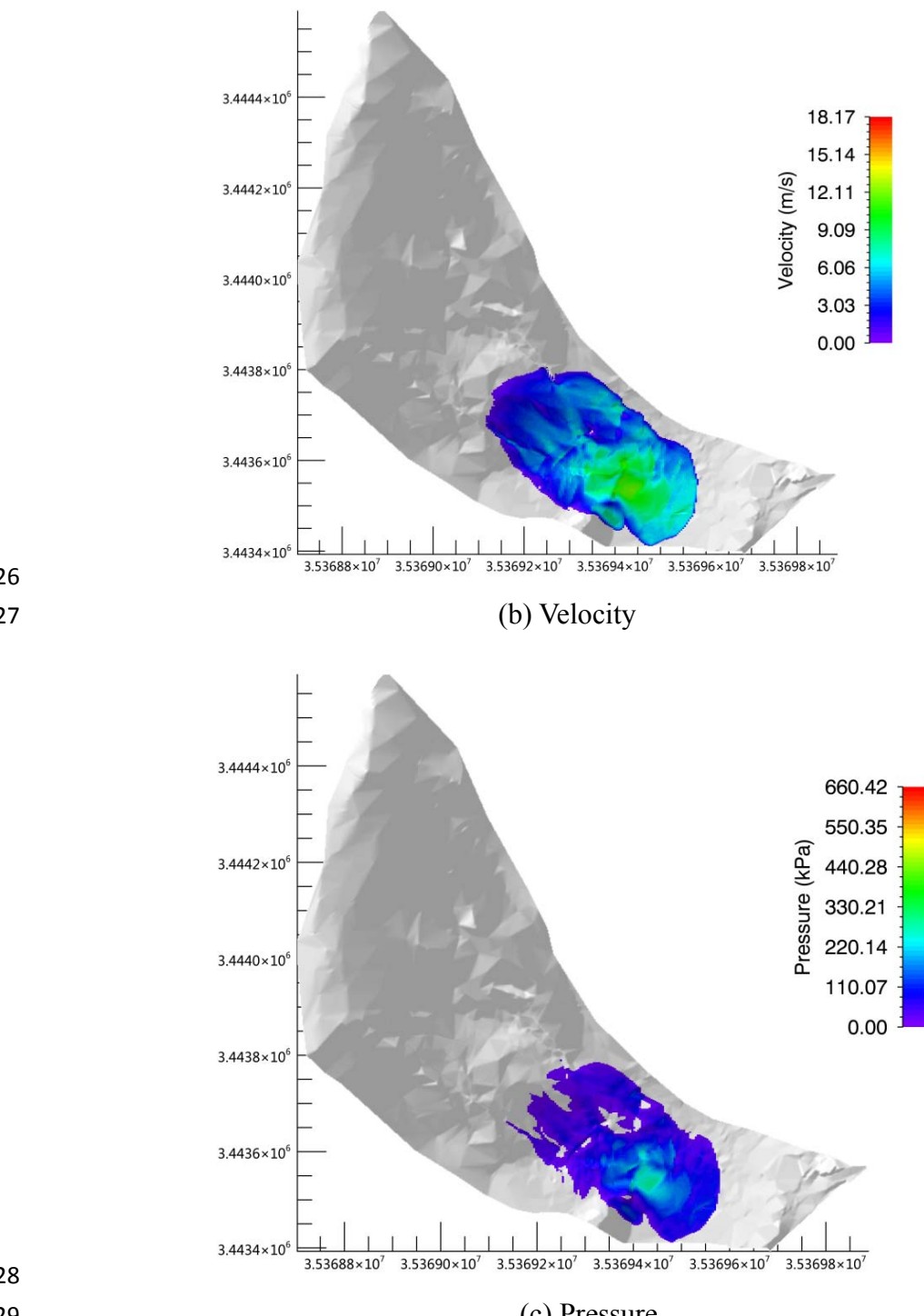


(b) Velocity


(c) Pressure

Fig. 9 Movement characteristic parameters of the Taziping landslide (after treatment)

Provided in Fig.9 are the kinematic characteristics of the landslide deposit. The
colored bar shows the maximum values of the kinematic process for a given time step.
Deposits accumulated during the landslide movement process had a maximum flow
height of 18.37m, located around the surface gully of the middle and upper slope. The
middle and lower portions of the landslide deposit had a flow height of approximately
3-5m. The middle and lower movement velocity of the landslide deposits ranged

between 3m/s and 5m/s. The landslide had a mean pressure of about 330kPa, and the pressure of the middle and lower deposits was about 100kPa. Thus, it could be held that two-story and lower houses within the deposition range might be buried. It was further suggested that the design strength of the gable walls of houses on the middle and upper parts of the deposits be increased above 150kPa.

After treatment, the accumulation flow height and pressure of the deposits were reduced by about 1/2, and the kinematic speed is reduced by about 1/3. However, the Miaoba residential area of Red Village was still partially at hazard.

**4 Results**

Landslides reflect landscape instability that evolves over meteorological and geological timescales, and they also pose threats to people, property, and the environment. The severity of these threats depends largely on landslide speed and travel distance. There may be examples where entire houses on a landslide mass are moved but not destroyed because of stable base plates. In any case, velocity plays a more important role regarding kinetic energy acting on an obstacle. However, the Miaoba residential area of Red Village is located at the frontal part of Tazhiping lanslide. During landslide movement, the spatial scale indexes of a landslide mass include area, volume, and thickness. The maximum thickness of the landslide is one of the direct factors influencing the building's deformation failure status. A large landslide displacement may lead to burial, collapse, or deformation failure of the building, and thus influence its safety and stability. Thus, landslide thickness constitutes an important index for assessing the hazards of a landslide disaster, and for influencing the consequences faced by disaster-affected bodies (Fell et al., 2008; DZ/T, 0286-2015). Provided in Tab.3 is a landslide thickness-based division of the predicted hazard zones of Taziping landslide, in which the thickness of the landslide mass correlates with the ability of a building to withstand a landslide disaster (Hungr et al., 1984; Petrazzuoli et al., 2004; Glade 2006; GB, 50010–2010; Hu et al., 2012; Zeng et al., 2015). After treatment with slide-resistant piles, the hazard of a future slide was reduced by about 1/3 overall and by 2/3 in high-hazard zones.

**Tab.3 Division table of the predicted hazards of Taziping landslide (unit: m$^2$)**

| Hazard zone level | Assessment index | Building damage probability | Area before treatment | Area after treatment | Increased/decreased area | Building damage characteristics |
|---|---|---|---|---|---|---|
| **Low-hazard zone (l)** | $h \leq 0.5m$ | 20% | 44,600 | 38,748 | -5,852 | One-story houses may be damaged; houses on the |

| | | | | | | |
|---|---|---|---|---|---|---|
| **Relatively low-hazard zone** (II) | 0.5 m < $h \leq$ 1m | 50~20% | 24,900 | 26,400 | +1,500 | landslide mass are partially damaged. One-story houses have a very high probability of being damaged; one-story houses on the landslide mass are completely damaged. |
| **Moderate-hazard zone** (III) | 1m < $h \leq$ 3m | 80~50% | 21,980 | 15,856 | -6,124 | One-story to three-story houses have a very high probability of being damaged; houses less than three stories on the landslide mass are completely damaged. |
| **Relatively high-hazard zone** (IV) | 3m < $h \leq$ 5m | 100~80% | 30,820 | 19,636 | -11,184 | One-story houses may be buried, and two-story to six-story houses have a very high probability of being damaged; houses on the landslide mass are completely |

| High-hazard zone (V) | $h \geq 5m$ | 100% | 47 , 240 | 13 , 052 | -34,188 | damaged. Two-story and lower houses may be buried, and three-story and higher houses have a very high probability of being damaged; houses on the landslide mass are completely damaged. |
|---|---|---|---|---|---|---|
| Total area: | — | — | 169 , 540 | 113 , 700 | -54,340 | — |

The hazard zones of Taziping landslide was given by 2D divisions before and
after engineering treatment (Fig. 10). The size of the hazard zones changed after
engineering treatment, particularly in the high-hazard zones. Before treatment with
slide-resistant piles, the landslide posed a great hazard to eight houses on the left side
of the upper Miaoba residential area, with a high-hazard zone associated with
landslide mass height over 5m and a red zone. After treatment, the number of effected
houses was reduced to four. We defined outside the colored area as no-hazard.

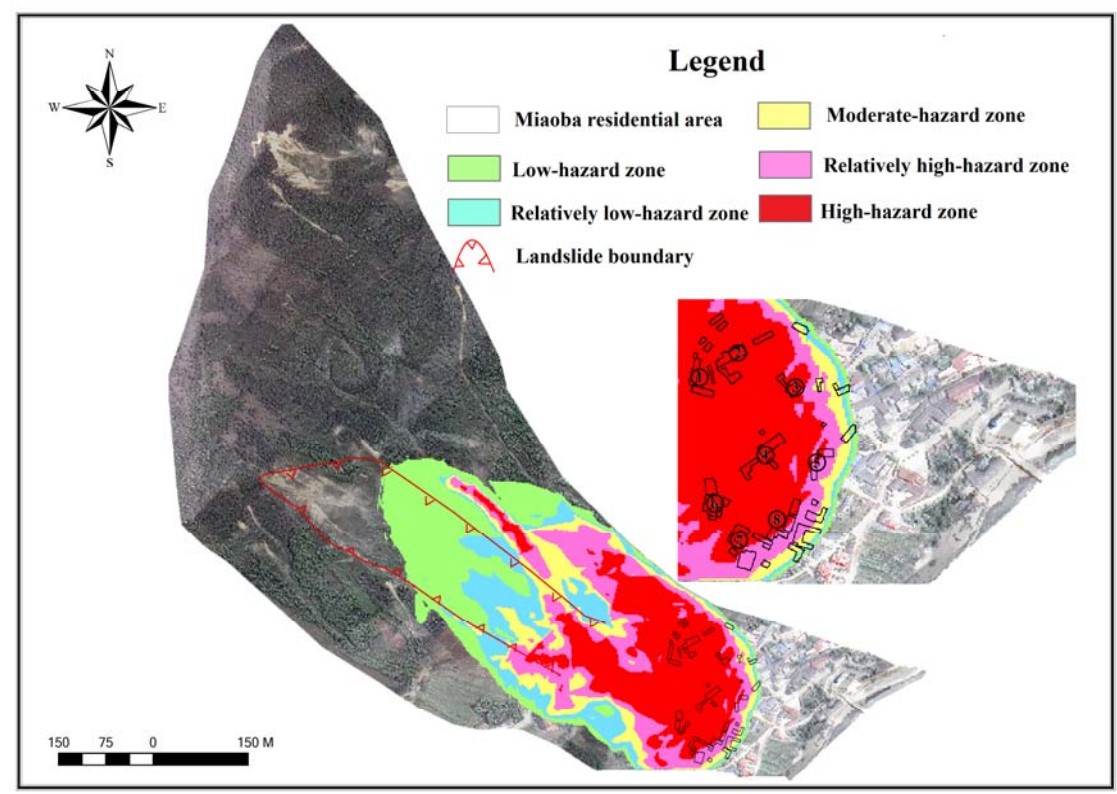


**(a) Before treatment**

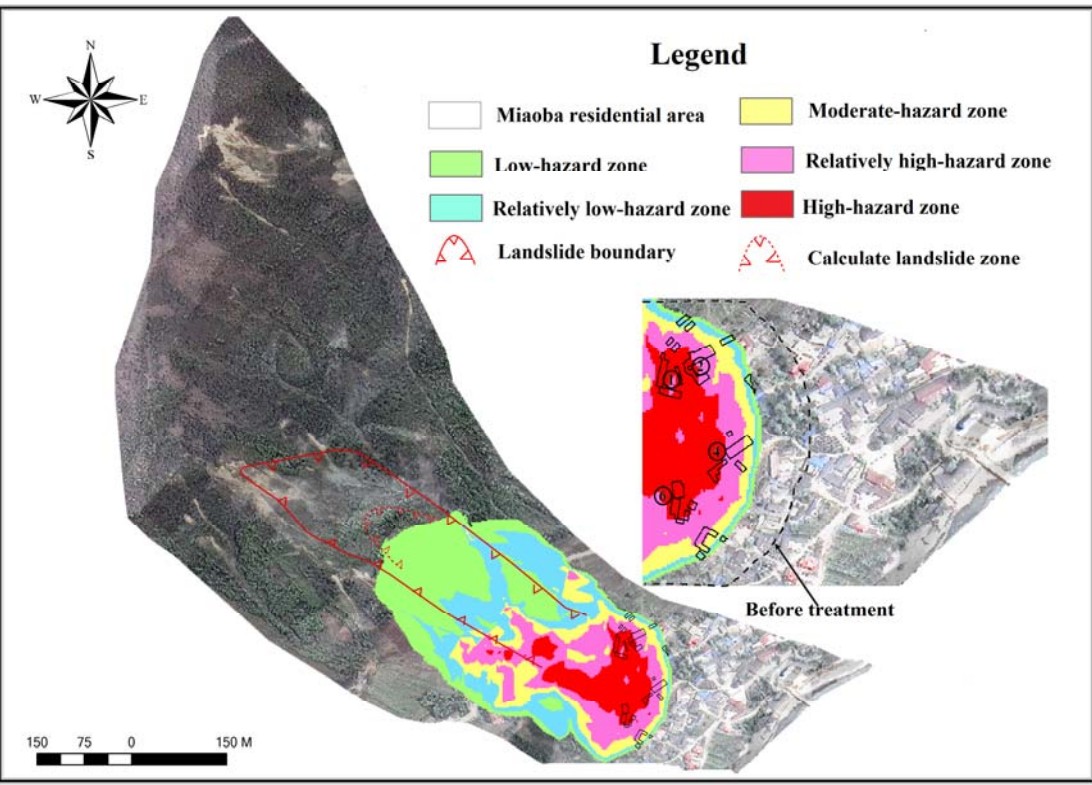


**(b) After treatment**
**Fig. 10 2D division comparison of the hazards of the Taziping landslide**

**5 Conclusions and Discussion**

The hazard assessment of landslides using numerical models is becoming more and more popular as new models are developed and become available for both scientific research and practical applications. There is some confusion about the mass movement process that is discussed by the rheological model presented in this contribution.

Landslides move downslope in many different ways (Varnes, 1978). In addition, landslides can evolve into rapidly travelling flows, which exhibit characteristics of debris flows on unchannelized or only weakly channelized hillslopes. The geomorphic heterogeneity of rapid shallow landslides, such as hillslope debris flows, is larger than observed in channelized debris flows; however many of these flows can be successfully modelled using the Voellmy-fluid friction (Christen et al., 2012). Results presented in this paper support the conclusion that Voellmy-fluid rheological model can be used to simulate flow-type landslides.

The selection of model parameters remains one of the fundamental challenges for numerical calculations of natural hazards. At present, there are numerous empirical parameters obtained from 30-years of monitoring data. Such as in RAMMS, we can automatically generate the friction coefficient of an avalanche for our calculation domain based on topographic data analysis, forest information and global parameters (WSL, 2013). The friction parameters for debris flows can found in some literature (Fannin et al., 2001; Iovine et al., 2003; Hürlimann et al., 2008; Scheidl et al., 2010; Huang et al., 2015). However, there is little research regarding friction parameters of flow-type landslide. Therefore, we tested different coulomb friction

coefficient $\mu$ values ranging between $0.1 \leq \mu \leq 0.6$ and viscous friction coefficient $\zeta$

values ranging between $100 \leq \mu \leq 1000 m \cdot s^{-2}$. Finally, we selected the coulomb

friction coefficient $\mu = 0.45$ and viscous friction coefficient $\zeta = 500 m \cdot s^{-2}$ in

accordance with back-analyses of well-documented landslides (Cepeda et al., 2010; Du et al., 2015). Simulation results are consistent with field observations of topography and sliding path.

Based on the finite volume method and the RAMMS program, simulation results of Taziping landslide were consistent with the sliding path predicted by the field investigation. This correlation indicates that numerical simulation is an effective method for studying the movement processes of flow-type landslides. The accumulation flow height and pressure of landslide deposits were reduced by about 1/2, and the kinematic speed was reduced by about 1/3 after treatment. However, the Miaoba residential area of Red Village is still partially at hazard. Considering that two-story and lower houses within the deposition range might be buried, it was further suggested that the design strength of the gable walls of houses on the middle and upper parts of the deposit be increased above 150kPa.

By utilizing a GIS platform in combination with landslide hazard assessment indexes, we mapped the 2D division of the Taziping landslide hazard zones before

and after engineering treatment. The results indicated that overall hazard zones contracted after engineering treatment and, the area of high-hazard zones was reduced by about 2/3. After engineering treatment, the number of at hazard houses on the left side of the upper Miaoba residential area, was reduced from eight to four. It was thus clear that some zones are still at high hazard despite engineering treatment. Therefore, it was proposed that houses located in high-hazard zones be relocated or reinforced for protection.

## Acknowledgments

The authors sincerely acknowledge the CAS Pioneer Hundred 432 Talents Program for the completion of this research. This work was supported by National Natural Science Foundation of China (Grant No. 41301009 41301592) and the Hundred Young Talents Program of IMHE (SDSQB-2016-01), the International Cooperation Program of the Ministry of Science and Technology of China (Grant No.2013DFA21720). The authors express their deepest gratitude to those aids and assistances. The authors also extend their gratitude to editor and two anonymous reviewers for their helpful suggestions and insightful comments, which have contributed greatly in improving the quality of the manuscript.

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
