# Peer review of "Hazard Assessment Comparison of Tazhiping Landslide Before and After Treatment using the Finite Volume Method"

_Natural Hazards and Earth System Sciences, 2016_

## Referee Comment (RC1) · Anonymous Referee #1 · 8 Mar 2017

The paper by Huang et al. addresses relevant scientific and technical questions. It presents a concept and adoption of a well-known method to simulate mass movement processes. The used methods are in principle up to international standards but there is some doubt whether they used the appropriate method for this study. The scientific methods and assumptions used are valid and outlined clearly. There is some confusion about the mass movement process that is discussed and approached by the presented and adopted rheological model. In principle, the numerical approach in RAMMS can also be used for the simulation of landslides. But it is actually not intended for it and does not take into account specific properties of this kind of mass movement (landslides). The results of the study are not really surprising. Interpretation of the simulation results is derived poorly. Some important questions remain still unanswered, namely the sensitivity of the friction parameters and - more important - the derivation

of the best-fit parameters presented in Table 2. This aspect should be at least considered in the discussion and ideally in the methods section. While the methods section is very detailed (and also well written in good English) regarding the numeric, no information is given about the modeling procedure and interpretation of the simulation results. The title does not promise detailed information about the numeric but rather a specification about the hazard assessment comparison. Therefore or the title or the content of the paper should be changed. The same is true for the abstract. More information should be given for the methods section or the method section should be adjusted. The mathematical formulae, symbols, abbreviations and units are correctly defined and used. There is some confusion in terminology for figures 6 and 7, that have to be changed. Figures should be improved. Figure 1 seems to be taken from an existing paper without citation. Figure 2 needs more information about the location of the study site in a global perspective and better visualization of the exact location in the Baisha river basin. figures 6 and 7 do not contain more details on the landslide area, location of the objects at risk, etc. This information is only given in figure 8 but visualized rather small. Readability of the outlines of buildings is very hard and not mentioned in the legend. The authors give in principle proper credit to previous and related work. Own contributions are not well indicated (besides the adoption of the model and the interpretation of the simulation results). Number and quality of the references are appropriate. There are some publications in Chinese that are not accessible by all fellow scientist. There is some confusion for the article by Zhang,Z.Y., Wang,S.T., Wang,L.S.,et al., about the year of publication. In the text 1994 is mentioned while in the references there is written 1993. The reference of Toro, 1992 is missing. Structure and length of the paper is adequate. Methods section with the numeric is too long compared to the results section. Technical language and the English is more or less of good quality and understandable. Several sentences need to be reformulated, mostly because of wrong word order. There is no supplementary material available.

p.2, line 61: what do the autors exactly mean with "landslide-debris flows?" Please rely on some definitions in the literature.

p.2, line 71: what to the autors exactly mean with 3D mapping of the division of hazard zones? Usually, hazards zonation is given on a map, e.g. in 2D

p.3, line 98: this figure is taken from Christen et al., 2010. Please cite source.

p.3, line 107: missing space

p.7, line 178: this reference is missing in the reference section

p.11, line 255: see comment for p.2, line 71

p.11, line 266: figure is subtitled with "Thickness". Thickness of deposition is not equal to flow height (if a landslide really "flows"...). Please adapt wording

p.12, line 268: subtitle of figure is "Speed", legend says "Velocity". If the blue to green marked zone shows the deposited mass of the landslide, there should be no velocity value (because it's deposited). In chapter 3 is no indication or estimation about the speed of the landslide mass, therefore figure 6b does not really make sense.

p.12, line 270: not clear, if the colored area shows the maximum pressure or an instantaneous for a given time step. Much more of interest would be a local value (over time) at the position of a building. And why the legend goes up to more than 1000kPa but no reddih or yellowish areas are marked?

p.12, lines 274, 277 and p.13, line 278: not clear what numbers in the circle mean. Is this kind of a list or does it indicate a location in a figure?

p.13, line 279: how is made this separation between houses of different numbers of stories? Please give more information and references to it.

p.13, line 293: or indicate "about 1.2 m" or give exact value

p.13, line 298: same remark as for figure 6a

p. 14, line 300: same remark as for figure 6b

p.14, line 305: example of a sentence that has to be rewritten because of wrong word
order

p.14, lines 305, 308, 309: not clear what numbers in the circle mean.

p.15, line 321/322: not sure, if this statement is really true. There may be examples where entire houses on a landslide mass are moved but not destroyed because of stable baseplates. In any case, velocity plays a more important role regarding kinetic energy acting on an obstacle. You are right in the sense that the height of a moving landslide (e.g. the frontal part) plays an important role when it hits a building on a higher level, e.g. the second or third floor. Please clarify this point.

p.15, 16 and 17, table 3: the term "washed away" is not suitable for landslide process. It implies an major influence by a fluid.

p.17, line 333: This should be 2D, because you show a map with the different zonations. These different zonations are not defined, by the way.

p.17, line 339: There seem to be marked buildings (in the red high-hazard zone). If so, adjust legend and make sure they are better visible. What zone is definde outside the colored area? No hazard or also low-hazard zone?

p.18, line 342: same as for figure 8a. And this should be 8b instead of 8c

p.18, line 350: what is a landslide-debris flow?

p.18, line 358: this should be 2D

p.19, line 411: correct reference would be: Michael-Leiba, M., Baynes, F., Scott, G., Granger, K. 2003. Regional landsliderisk to the Cairns community [J]. NatHazards, 2003,30 (2):233–249. Check reference style for all references according to the journal style!

---

## Referee Comment (RC2) · Anonymous Referee #2 · 14 Mar 2017

The manuscript presented a fluid mechanics based method for landslide/debris flow modeling, and was further applied to a real landslide case for hazard zones mapping. The topic is scientifically significant for nature hazard mitigations. The manuscript was logically organized and the results were well described and reasonably discussed. The authors provided sufficient evidence that the proposed method could be used as a promising tool for landslide modelling and hazard mapping. The knowledge obtained from the study would benefit civil engineering society for landslide investigation assessment. This paper can be accepted for publication by considering all the points given below.

1. The main contribution of this paper seems to be the computational model proposed. It is desired to add related descriptions to the title of this paper.

[Figure]

2. Previous study on landslide/debris flow issues using the fluid mechanics based method had faced the problem that it predicts higher mobility of the moving body while using the same fluid parameters throughout the whole flowing process. For example, less obvious fluid property is expected when the flow body is approaching stop point. It is stated in this manuscript that a changed frictional resistance is used (L78). However, the details are not clear in the text. Relevant descriptions on this issue should be strengthened.

3. It is not clear in the text that how the free surface of the landslide/debris flow is treated or reconstructed. An additional figure is need to describe the details.

4. Fig.4 showed the geological profile of Taziping Landslide and a slide surface is clearly indicated. Is this slide surface comparable with the simulation result? It would be interesting to show their comparison.

5. In Tab.3, Various hazard zone levels were cataloged. What is the criterial to assign a specific damage situation to a certain zone level? Is there any standard code to follow?

Other specific comments are given below. 1. The quotations in the manuscript are not in the same format, for example, Line 44, Costa, 1984; VS Line 50, Zhang. Y, 2013. Usually only family name is preferred, please refer to the journal's instructions and make necessary changes throughout the text.

2. Fig.1 needs proper citation.

3. In Fig.6, Fig.7, what moment of flow does these figures represent? Different moment should have different deposit thickness, flow velocity and pressure. Please confirm.

4. L276 "The middle and lower deposits had a thickness of 277 about 5-10m", confusing here, what does "the middle and lower deposits" mean? Similar as "the middle and lower movement speed", please check throughout the text.

5. L289. What technique is used for searching the sliding plane?

6. L305, Fig.4 should be Fig.7.

7. Tab.3. How is the "Building damage probability" evaluated?

---

## Author Comment (AC1) · 28 Apr 2017

Manuscript title: (the original title: Hazard Assessment Comparison of Tazhiping Landslide Before and After Treatment) Manuscript number: 2016-391 Thanks very much for reviewer's comments, which helped us to improve the quality of manuscript. We have made a major revision to address all the comments raised by the reviewer. All changes have been marked with RED color in the revised manuscript. We would be happy to make further modifications if required. We hope the changes listed have made the manuscript suitable for publication and we look forward to your response.

Q1: Some important questions remain still unanswered, namely the sensitivity of the friction parameters and more important the derivation of the best-fit parameters presented in Table 2. This aspect should be at least considered in the discussion and ide-

ally in the methods section. A1: It is an important issue on the derivation of the best-fit calculated parameters, and we have considered in the discussion and methodology sections. The present estimation of model parameters can be acquired by laboratory or small-scale experiments in some instance, however the Voellmy rheological model friction coefficient generally lacks a systematic approach to get. Therefore, we tested different coulomb friction coefficient values ranging between and viscous friction coefficient values ranging between . Finally, we selected the coulomb friction coefficient and viscous friction coefficient in accordance with back-analyses of well-documented landslide cases (Cepeda, J., et al. 2010; Du et al., 2015 ). The text in the method section and discussion section have been revised. Please see p.12, line 267-274 and p.23, line 389-440.

Q2: The title does not promise detailed information about the numeric but rather a specification about the hazard assessment comparison. Therefore or the title or the content of the paper should be changed. The same is true for the abstract. A2: The title of this paper has been revised to "Hazard Assessment Comparison of Tazhiping Landslide Before and After Treatment Using Finite Volume Method". The corresponding abstract has been revised as well. Please see p.1, line 2 and line 12-13.

Q3: There is some confusion in terminology for figures 6 and 7, that have to be changed. Figures should be improved. Figure 1 seems to be taken from an existing paper without citation.Figure 2 needs more information about the location of the study site in a global perspective and better visualization of the exact location in the Baisha river basin. figures 6 and 7 do not contain more details on the landslide area, location of the objects at risk, etc. This information is only given in figure 8 but visualized rather small. Readability of the outlines of buildings is very hard and not mentioned in the legend. A3: The confusion in terminology for Figures.6 and 7 have been revised. Please see p.13,line 278; p.14,line 280; p.16,line 317 and p.17,line 319. We have re-organized and added more information about the location of the studying site and Baisha river basin was shown in Figure 2. Please see p.9,line 212-214. In Figures 6 and 7 we add more

details on the landslide area shown in Figures 7a and 9a. Please see p.13,line 277-278. Figure 10 has been extensively visualized and added the outlines of buildings in the legend. Please see p.21-22, line 367-373. Various minor modification and revision were made in all Figures.

Q4: There are some publications in Chinese that are not accessible by all fellow scientist. There is some confusion for the article by Zhang,Z.Y., Wang,S.T., Wang,L.S.,et al., about the year of publication. In the text 1994 is mentioned while in the references there is written 1993. The reference of Toro, 1992 is missing. A4: We have deleted some parts of unimportant Chineses literature and revised all references according to the NHESSD journal style. We have cited the reference of Toro, 1992. Please see references section.

Other specific comments are given below. Q5: p.2, line 61: what do the authors exactly mean with "landslide-debris flows?" Please rely on some definitions in the literature. A5: Landslides move downslope in many different ways (Varnes, 1978).Flow-type landslides can evolve into rapidly travelling flows, which exhibit characteristics of debris flows on unchannelized or only weakly channelized hillslopes. The geomorphic heterogeneity of rapid shallow flow-type landslides such as hillslope debris flows is larger than those observed in channelized debris flows, however, many of these flows can be successfully modelled using the Voellmy-fluid friction relation and initiating the flow as a block release (Christen et al., 2012 ). It is true that there is some confusion about the term "landslide-debris flows" we used here. We have revised it to "flow-type landslides" and add some definitions in the literature. Please see p2, line 63-64 and discussion section.

Q6: p.2, line 71: what to the autors exactly mean with 3D mapping of the division of hazard zones? Usually, hazards zonation is given on a map, e.g. in 2D A6: It has been revised to 2D. Please see p.2, line 74.

Q7: p.3, line 98: this figure is taken from Christen et al., 2010. Please cite source. A7:It

has been added. Please see p.3, line 101.

Q8: p.3, line 107: missing space. A8: It has been revised. Please see p.4, line 109.

Q9: p.7, line 178: this reference is missing in the reference section. A9: It has been cited. Please see reference p.26, line 530-531. " Toro, E.F.: Riemann problems and the waf method for solving the two dimensional shallow water equations, Philos. Trans. R. Soc. London, Ser., A 338, 43–68. 1992".

Q10: p.11, line 255: see comment for p.2, line 71 A10: It has been revised to 2D. Please see p.12, line 259.

Q11: p.11, line 266: figure is subtitled with "Thickness". Thickness of deposition is not equal to flow height (if a landslide really "flows"...). Please adapt wording. A11: It has been revised to flow height. Please see p.1, line 15 ;see p.13, line 278 ; p14, line 287; p15, line 289; p.16, line 317; p17, line 327 and 328; p18, line 336; p23, line 411.

Q12: p.12, line 268: subtitle of figure is "Speed", legend says "Velocity". If the blue to green marked zone shows the deposited mass of the landslide, there should be no velocity value (because it's deposited). In chapter 3 is no indication or estimation about the speed of the landslide mass, therefore figure 6b does not really make sense. A12: It has been revised to Velocity. Please see p.1, line 15 ;p.13, line 280; p15, line 290 and p.17, line 329.In any case, velocity plays a more important role regarding kinetic energy acting on an obstacle. However, the Miaoba residential area of Red Village is located at the frontal part of Tazhiping lanslide. Therefore, the maximum flow height of the landslide is one of the direct factors influencing the building's deformation failure status. Please see p.18, line 339-348 and p17, line 329.

Q13: p.12, line 270: not clear, if the colored area shows the maximum pressure or an instantaneous for a given time step. Much more of interest would be a local value (over time) at the position of a building. And why the legend goes up to more than 1000kPa but no reddih or yellowish areas are marked? A13: The coloredbar shows the

maximum values of moving process or an instantaneous pressure for a given time step. As the building of Red Village is located at the frontal part of landslide, the pressure of the middle and lower landslide deposits was about 200kPa. Thus, three-story and lower houses within the deposition range might be buried. The maximum pressure value in the surface gully can be found in the middle and upper slope. According to field survey we have found this gully is in the elevation of about 1,200 m. The maximum pressure value is easy been found from the instantaneous for a given time step figures. Therefore, coupled with field observations and numerical simulation, they are especially helpful in understanding landslide movement process in complex terrain. It has been introduced in p.17, line 324-325.

Q14: p.12, lines 274, 277 and p.13, line 278: not clear what numbers in the circle mean. Is this kind of a list or does it indicate a location in a figure? A14: No, it does not indicate a location. It has been deleted. Please see p.18, line 339-348.

Q15: p.13, line 279: how is made this separation between houses of different numbers of stories? Please give more information and references to it. A15: The building is 3m height each floor in China. We have cited some literatures (Hungr et al., 1984; Petrazzuoli et al., 2004; GB, 50010–2010; Hu et al., 2012; Zeng et al., 2015). Please see p18, line 358 and 359.

Q16: p.13, line 293: or indicate "about 1.2 m" or give exact value. A16: The more exact value has been given . " with an elevation of 1,070-1,072m and a length of 182m." Please see p.15, line 312-313.

Q17: p.13, line 298: same remark as for figure 6a. A17: It has been revised. Please see A.11.

Q18: p. 14, line 300: same remark as for figure 6b A18: It has been revised. Please see A.12.

Q19: p.14, line 305: example of a sentence that has to be rewritten because of wrong

word order A19:We have revised to "Provided in Fig.7 are the kinematic characteristics of the landslide deposit." Please see p.17, line 324.

Q20: p.14, lines 305, 308, 309: not clear what numbers in the circle mean. A20: It has been deleted. Please see p.17, line 325-330.

Q21: p.15, line 321/322: not sure, if this statement is really true. There may be examples where entire houses on a landslide mass are moved but not destroyed because of stable base plates. In any case, velocity plays a more important role regarding kinetic energy acting on an obstacle. You are right in the sense that the height of a moving landslide (e.g. the frontal part) plays an important role when it hits a building on a higher level, e.g. the second or third floor. Please clarify this point. A21: We have clarified this point. "Landslides reflect landscape instability that evolves over meteorological and geological timescales, and they also pose threats to people, property, and the environment. The severity of these threats depends largely on landslide speed and travel distance. There may be examples where entire houses on a landslide mass are moved but not destroyed because of stable base plates. In any case, velocity plays a more important role regarding kinetic energy acting on an obstacle. However, the Miaoba residential area of Red Village is located at the frontal part of Tazhiping lanslide." Please see p.18,lines 341-348.

Q22: p.15, 16 and 17, table 3: the term "washed away" is not suitable for landslide process. It implies an major influence by a fluid. A22: It has been revised. Please see p.18,lines 361.

Q23: p.17, line 333: This should be 2D, because you show a map with the different zonations. These different zonations are not defined, by the way. A23: It has been revised to 2D. Please see p.20, line 362 and p.22,line 375.

Q24: p.17, line 339: There seem to be marked buildings (in the red high-hazard zone). If so, adjust legend and make sure they are better visible. What zone is defined outside the colored area? No hazard or also low-hazard zone? A24: We have adjusted legend

and defined outside the colored area as no-hazard. Please see p.20, line 366-368 and Figure.10 legend.

Q25: p.18, line 342: same as for figure 8a. And this should be 8b instead of 8c A25: It has been revised. Please see p.22, line 374.

Q26: p.18, line 350: what is a landslide-debris flow? A26: It has been defined. Please see p.23, line 410 and answer A5.

Q27: p.18, line 358: this should be 2D A27: It has been revised. Please see p.23, line 418.

Q28: p.19, line 411: correct reference would be: Michael-Leiba, M., Baynes, F" Scott, G., Granger, K. 2003. Regional landslide risk to the Cairns community [J]. NatHazards, 2003,30 (2):233–249. Check reference style for all references according to the journal style! A28: We have revised all references according to the NHESSD journal style. The reference list has been updated as well. Please see references section.

The text of the manuscript has been revised.

---

## Author Comment (AC2) · 28 Apr 2017

Manuscript title: (the original title: Hazard Assessment Comparison of Tazhiping Landslide Before and After Treatment) Manuscript number: 2019-391 Thanks very much for reviewer's comments, which helped us to improve the quality of manuscript. We have made a major revision to address all the comments raised by the reviewer. In the revised manuscript, all changes have been marked in RED which is suggested by the reviewers, and is modified by the authors. We would be happy to make further modifications if required. We hope the changes listed have made the manuscript suitable for publication and we look forward to your response.

Q1: The main contribution of this paper seems to be the computational model proposed. It is desired to add related descriptions to the title of this paper. A1: This paper

title has been revised to "Hazard Assessment Comparison of Tazhiping Landslide Before and After Treatment Using Finite Volume Method". Please see p.1, line 2.

Q2: Previous study on landslide/debris flow issues using the fluid mechanics based method had faced the problem that it predicts higher mobility of the moving body while using the same fluid parameters throughout the whole flowing process. For example, less obvious fluid property is expected when the flow body is approaching stop point. It is stated in this manuscript that a changed frictional resistance is used (L78). However, the details are not clear in the text. Relevant descriptions on this issue should be strengthened. A2: This paper adopted the RAMMS to simulate the mass movement process. In RAMMS, we can automatically generate the friction coefficient for our calculation domain based on topographic data analysis, forest information and global parameters and so on. Therefore, we can used a changed frictional resistance. This problem has considered in the discussion section. Please see p.22-23, line 378~406.

Q3: It is not clear in the text that how the free surface of the landslide/debris flow is treated or reconstructed. An additional figure is need to describe the details. A3: We have reconstructed and added Figure4. Please see p.10, line 218.

Q4: Fig.4 showed the geological profile of Taziping Landslide and a slide surface is clearly indicated. Is this slide surface comparable with the simulation result? It would be interesting to show their comparison. A4: We have reconstructed and added Figure8. Before engineering treatment, Figure.4 and Figure.5 have showed that the sliding mass had an estimated starting volume of about 600,000m3 and a mean thickness of 8m. After fully accounting for the slide-resistant piles and mounds, we introduced the Morgenstern-Price method to calculate the stability coefficient of Taziping landslide after treatment. The method was determined with an iterative approaching by changing the position of the sliding surface until failure of the dumpsite (Figure.8). Please see p.15, line 300~302 and 306~308.

Q5: In Tab.3, Various hazard zone levels were cataloged. What is the criterial to assign

a specific damage situation to a certain zone level? Is there any standard code to follow? A5: We have cited standard code and literature( Fell R et al., 2008; DZ/T 0286-2015). Please see p.18, line 354∼355.

Other specific comments are given below.

Q6: The quotations in the manuscript are not in the same format, for example, Line 44, Costa, 1984; VS Line 50, Zhang. Y, 2013. Usually only family name is preferred, please refer to the journal's instructions and make necessary changes throughout the text. p.11, line 266: figure is subtitled with A6:It has been revised. We have revised all references and quotations in the manuscript according to the NHESSD journal style. The reference list has been updated as well. Please see references and quotations section.

Q7: Fig.1 needs proper citation. A7: It has been revised (Christen et al., 2010a).

Q8: In Fig.6, Fig.7, what moment of flow does these figures represent? Different moment should have different deposit thickness, flow velocity and pressure. Please confirm. A8: The Figure.6 and Figure.7 is shown that the last moment of the flow. Different moment have different deposit flow height, velocity and pressure. However, the coloredbar shows the maximum values or an instantaneous for a given time step. It has been revised. Please see p.17, line 324-325.

Q9: L276 "The middle and lower deposits had a thickness of about 5-10m", confusing here, what does "the middle and lower deposits" mean? Similar as "the middle and lower movement speed", please check throughout the text. A9: This sentences has been reformulated, because of wrong word order. Please see p.14, line 288 and p.17, line 328.

Q10: L289. What technique is used for searching the sliding plane. A10: Coupled with field borehole surveying and numerical calculation method to search the sliding plane.

Q11: L305, Fig.4 should be Fig.7. A11: It has been revised. Please see p.17, line 324.

Please see p18, line 358 and 359.

Q12: Tab.3. How is the "Building damage probability" evaluated?. A12: By the thickness of the landslide mass to evaluate the ability of a building to withstand a landslide disaster. I have cited some literatures (Hungr et al., 1984; Petrazzuoli et al., 2004; GB, 50010–2010; Hu et al., 2012; Zeng et al., 2015). Please see p18, line 358 and 359.

The text of the manuscript has been revised.

---

## Author Comment (AC3) · 28 Apr 2017

Manuscript title: (the original title: Hazard Assessment Comparison of Tazhiping Landslide Before and After Treatment) Manuscript number: 2019-391 Thanks very much for reviewer's comments, which helped us to improve the quality of manuscript. We have made a major revision to address all the comments raised by the reviewer. All changes have been marked with RED color in the revised manuscript. We would be happy to make further modifications if required. We hope the changes listed have made the manuscript suitable for publication and we look forward to your response.

Q1: The main contribution of this paper seems to be the computational model proposed. It is desired to add related descriptions to the title of this paper.

A1: The title of this paper has been revised to "Hazard Assessment Comparison of

Tazhiping Landslide Before and After Treatment Using Finite Volume Method". Please see p.1, line 2.

Q2: Previous study on landslide/debris flow issues using the fluid mechanics based method had faced the problem that it predicts higher mobility of the moving body while using the same fluid parameters throughout the whole flowing process. For example, less obvious fluid property is expected when the flow body is approaching stop point. It is stated in this manuscript that a changed frictional resistance is used (L78). However, the details are not clear in the text. Relevant descriptions on this issue should be strengthened.

A2: This paper adopted the RAMMS to simulate the mass movement process. In RAMMS, we can automatically generate the friction coefficient for our calculation domain based on topographic data analysis, forest information and global parameters and so on. Therefore, we can use a changed frictional resistance. This problem has considered in the discussion section. Please see p.22-23, line 378~406.

Q3: It is not clear in the text that how the free surface of the landslide/debris flow is treated or reconstructed. An additional figure is need to describe the details.

A3: We have reconstructed and added Figure4. Please see p.10, line 218.

Q4: Fig.4 showed the geological profile of Taziping Landslide and a slide surface is clearly indicated. Is this slide surface comparable with the simulation result? It would be interesting to show their comparison.

A4: We have reconstructed and added Figure8. Before engineering treatment, Figure.4 and Figure.5 have showed that the sliding mass had an estimated starting volume of about 600,000m3 and a mean thickness of 8m. After fully accounting for the slide-resistant piles and mounds, we introduced the Morgenstern-Price method to calculate the stability coefficient of Taziping landslide after treatment. The method was determined with an iterative approaching by changing the position of the sliding surface until

failure of the dumpsite (Figure.8). Please see p.15, line 300~302 and 306~308.

Q5: In Tab.3, Various hazard zone levels were cataloged. What is the criterial to assign a specific damage situation to a certain zone level? Is there any standard code to follow?

A5: We have cited standard code and literature( Fell R et al., 2008; Qiao , 2009; DZ/T 0286-2015). Please see p.18, line 354~355.

Other specific comments are given below.

Q6: The quotations in the manuscript are not in the same format, for example, Line 44, Costa, 1984; VS Line 50, Zhang. Y, 2013. Usually only family name is preferred, please refer to the journal's instructions and make necessary changes throughout the text. p.11, line 266: figure is subtitled with.

A6:It has been revised. We have revised all references and quotations in the manuscript according to the NHESSD journal style. The reference list has been updated as well. Please see references and quotations section.

Q7: Fig.1 needs proper citation. A7: It has been revised (Christen et al., 2010a).

Q8: In Fig.6, Fig.7, what moment of flow does these figures represent? Different moment should have different deposit thickness, flow velocity and pressure. Please confirm.

A8: The Figure.6 and Figure.7 is shown that the last moment of the flow. Different moment have different deposit flow height, velocity and pressure. However, the coloredbar shows the maximum values of mowing process or an instantaneous for a given time step. It has been revised. Please see p.17, line 324-325.

Q9: L276 "The middle and lower deposits had a thickness of about 5-10m", confusing here, what does "the middle and lower deposits" mean? Similar as "the middle and lower movement speed", please check throughout the text.

[Figure]

A9: This sentences has been reformulated, because of wrong word order. Please see p.14, line 288 and p.17, line 328.

Q10: L289. What technique is used for searching the sliding plane.

A10: The method coupled with field borehole surveying and numerical calculation method described in Q4 were used to search the sliding plane.

Q11: L305, Fig.4 should be Fig.7.

A11: It has been revised. Please see p.17, line 324. Please see p18, line 358 and 359.

Q12: Tab.3. How is the "Building damage probability" evaluated?.

A12: By the thickness of the landslide mass to evaluate the ability of a building to withstand a landslide disaster. We have cited relevant literatures (Hungr et al., 1984; Petrazzuoli et al., 2004; GB, 50010–2010; Hu et al., 2012; Zeng et al., 2015). Please see p18, line 358 and 359.

The text of the manuscript has been revised.

---

## Author Comment (AC4) · 28 Apr 2017

Manuscript title: (the original title: Hazard Assessment Comparison of Tazhiping Landslide Before and After Treatment) Manuscript number: 2016-391 Thanks very much for reviewer's comments, which helped us to improve the quality of manuscript. We have made a major revision to address all the comments raised by the reviewer. All changes have been marked with RED color in the revised manuscript. We would be happy to make further modifications if required. We hope the changes listed have made the manuscript suitable for publication and we look forward to your response.

Q1: Some important questions remain still unanswered, namely the sensitivity of the friction parameters and more important the derivation of the best-fit parameters presented in Table 2. This aspect should be at least considered in the discussion and ideally in the methods section.

A1: It is an important issue on the derivation of the best-fit calculated parameters, and we have considered in the discussion and methodology sections. The present estimation of model parameters can be acquired by laboratory or small-scale experiments in some instance, however the Voellmy rheological model friction coefficient generally lacks a systematic approach to get. Therefore, we tested different coulomb friction coefficient values ranging between and viscous friction coefficient values ranging between . Finally, we selected the coulomb friction coefficient and viscous friction coefficient in accordance with back-analyses of well-documented landslide cases (Cepeda, J., et al. 2010; Du et al., 2015 ). The text in the method section and discussion section have been revised. Please see p.12, line 267-274 and p.23, line 389-440.

Q2: The title does not promise detailed information about the numeric but rather a specification about the hazard assessment comparison. Therefore or the title or the content of the paper should be changed. The same is true for the abstract.

A2: The title of this paper has been revised to "Hazard Assessment Comparison of Tazhiping Landslide Before and After Treatment Using Finite Volume Method". The corresponding abstract has been revised as well. Please see p.1, line 2 and line 12-13.

Q3: There is some confusion in terminology for figures 6 and 7, that have to be changed. Figures should be improved. Figure 1 seems to be taken from an existing paper without citation.Figure 2 needs more information about the location of the study site in a global perspective and better visualization of the exact location in the Baisha river basin. figures 6 and 7 do not contain more details on the landslide area, location of the objects at risk, etc. This information is only given in figure 8 but visualized rather small. Readability of the outlines of buildings is very hard and not mentioned in the legend.

A3: The confusion in terminology for Figures.6 and 7 have been revised. Please see p.13,line 278; p.14,line 280; p.16,line 317 and p.17,line 319. We have re-organized and added more information about the location of the studying site and Baisha river basin was shown in Figure 2. Please see p.9,line 212-214. In Figures 6 and 7 we add more details on the landslide area shown in Figures 7a and 9a. Please see p.13,line 277-278. Figure 10 has been extensively visualized and added the outlines of buildings in the legend. Please see p.21-22, line 367-373. Various minor modification and revision were made in all Figures.

Q4: There are some publications in Chinese that are not accessible by all fellow scientist. There is some confusion for the article by Zhang,Z.Y., Wang,S.T., Wang,L.S.,et al., about the year of publication. In the text 1994 is mentioned while in the references there is written 1993. The reference of Toro, 1992 is missing.

A4: We have deleted some parts of unimportant Chineses literature and revised all references according to the NHESSD journal style. We have cited the reference of Toro, 1992. Please see references section.

Other specific comments are given below.

Q5: p.2, line 61: what do the authors exactly mean with "landslide-debris flows?" Please rely on some definitions in the literature.

A5: Landslides move downslope in many different ways (Varnes, 1978).Flow-type landslides can evolve into rapidly travelling flows, which exhibit characteristics of debris flows on unchannelized or only weakly channelized hillslopes. The geomorphic heterogeneity of rapid shallow flow-type landslides such as hillslope debris flows is larger than those observed in channelized debris flows, however, many of these flows can be successfully modelled using the Voellmy-fluid friction relation and initiating the flow as a block release (Christen et al., 2012 ). It is true that there is some confusion about the term "landslide-debris flows" we used here. We have revised it to "flow-type landslides" and add some definitions in the literature. Please see p2, line 63-64 and discussion section.

Q6: p.2, line 71: what to the autors exactly mean with 3D mapping of the division of hazard zones? Usually, hazards zonation is given on a map, e.g. in 2D.

A6: It has been revised to 2D. Please see p.2, line 74.

Q7: p.3, line 98: this figure is taken from Christen et al., 2010. Please cite source.

A7:It has been added. Please see p.3, line 101.

Q8: p.3, line 107: missing space.

A8: It has been revised. Please see p.4, line 109.

Q9: p.7, line 178: this reference is missing in the reference section.

A9: It has been cited. Please see reference p.26, line 530-531. " Toro, E.F.: Riemann problems and the waf method for solving the two dimensional shallow water equations, Philos. Trans. R. Soc. London, Ser., A 338, 43–68. 1992".

Q10: p.11, line 255: see comment for p.2, line 71

A10: It has been revised to 2D. Please see p.12, line 259. Q11: p.11, line 266: figure is subtitled with "Thickness". Thickness of deposition is not equal to flow height (if a landslide really "flows"...). Please adapt wording. A11: It has been revised to flow height. Please see p.1, line 15 ;see p.13, line 278 ; p14, line 287; p15, line 289; p.16, line 317; p17, line 327 and 328; p18, line 336; p23, line 411.

Q12: p.12, line 268: subtitle of figure is "Speed", legend says "Velocity". If the blue to green marked zone shows the deposited mass of the landslide, there should be no velocity value (because it's deposited). In chapter 3 is no indication or estimation about the speed of the landslide mass, therefore figure 6b does not really make sense.

A12: It has been revised to Velocity. Please see p.1, line 15 ;p.13, line 280; p15, line 290 and p.17, line 329.In any case, velocity plays a more important role regarding kinetic energy acting on an obstacle. However, the Miaoba residential area of Red

Village is located at the frontal part of Tazhiping lanslide. Therefore, the maximum flow height of the landslide is one of the direct factors influencing the building's deformation failure status. Please see p.18, line 339-348 and p17, line 329.

Q13: p.12, line 270: not clear, if the colored area shows the maximum pressure or an instantaneous for a given time step. Much more of interest would be a local value (over time) at the position of a building. And why the legend goes up to more than 1000kPa but no reddih or yellowish areas are marked?

A13: The coloredbar shows the maximum values of moving process or an instantaneous pressure for a given time step. As the building of Red Village is located at the frontal part of landslide, the pressure of the middle and lower landslide deposits was about 200kPa. Thus, three-story and lower houses within the deposition range might be buried. The maximum pressure value in the surface gully can be found in the middle and upper slope. According to field survey we have found this gully is in the elevation of about 1,200 m. The maximum pressure value is easy been found from the instantaneous for a given time step figures. Therefore, coupled with field observations and numerical simulation, they are especially helpful in understanding landslide movement process in complex terrain. It has been introduced in p.17, line 324-325.

Q14: p.12, lines 274, 277 and p.13, line 278: not clear what numbers in the circle mean. Is this kind of a list or does it indicate a location in a figure?

A14: No, it does not indicate a location. It has been deleted. Please see p.18, line 339-348.

Q15: p.13, line 279: how is made this separation between houses of different numbers of stories? Please give more information and references to it.

A15: The building is 3m height each floor in China. We have cited some literatures (Hungr et al., 1984; Petrazzuoli et al., 2004; GB, 50010–2010; Hu et al., 2012; Zeng et al., 2015). Please see p18, line 358 and 359.

Q16: p.13, line 293: or indicate "about 1.2 m" or give exact value.

A16: The more exact value has been given . " with an elevation of 1,070-1,072m and a length of 182m." Please see p.15, line 312-313.

Q17: p.13, line 298: same remark as for figure 6a.

A17: It has been revised. Please see A.11.

Q18: p. 14, line 300: same remark as for figure 6b .

A18: It has been revised. Please see A.12.

Q19: p.14, line 305: example of a sentence that has to be rewritten because of wrong word order.

A19:We have revised to "Provided in Fig.7 are the kinematic characteristics of the landslide deposit." Please see p.17, line 324.

Q20: p.14, lines 305, 308, 309: not clear what numbers in the circle mean.

A20: It has been deleted. Please see p.17, line 325-330.

Q21: p.15, line 321/322: not sure, if this statement is really true. There may be examples where entire houses on a landslide mass are moved but not destroyed because of stable base plates. In any case, velocity plays a more important role regarding kinetic energy acting on an obstacle. You are right in the sense that the height of a moving landslide (e.g. the frontal part) plays an important role when it hits a building on a higher level, e.g. the second or third floor. Please clarify this point.

A21: We have clarified this point. "Landslides reflect landscape instability that evolves over meteorological and geological timescales, and they also pose threats to people, property, and the environment. The severity of these threats depends largely on landslide speed and travel distance. There may be examples where entire houses on a landslide mass are moved but not destroyed because of stable base plates. In any

case, velocity plays a more important role regarding kinetic energy acting on an obstacle. However, the Miaoba residential area of Red Village is located at the frontal part of Tazhiping lanslide." Please see p.18,lines 341-348.

Q22: p.15, 16 and 17, table 3: the term "washed away" is not suitable for landslide process. It implies an major influence by a fluid.

A22: It has been revised. Please see p.18,lines 361.

Q23: p.17, line 333: This should be 2D, because you show a map with the different zonations.These different zonations are not defined, by the way.

A23: It has been revised to 2D. Please see p.20, line 362 and p.22,line 375.

Q24: p.17, line 339: There seem to be marked buildings (in the red high-hazard zone). If so,adjust legend and make sure they are better visible. What zone is defined outside the colored area? No hazard or also low-hazard zone?

A24: We have adjusted legend and defined outside the colored area as no-hazard. Please see p.20, line 366-368 and Figure.10 legend.

Q25: p.18, line 342: same as for figure 8a. And this should be 8b instead of 8c.

A25: It has been revised. Please see p.22, line 374.

Q26: p.18, line 350: what is a landslide-debris flow?

A26: It has been defined. Please see p.23, line 410 and answer A5.

Q27: p.18, line 358: this should be 2D.

A27: It has been revised. Please see p.23, line 418.

Q28: p.19, line 411: correct reference would be: Michael-Leiba, M., Baynes, F" Scott, G.,Granger, K. 2003. Regional landslide risk to the Cairns community [J]. NatHazards, 2003,30 (2):233–249. Check reference style for all references according to the journal style!

A28: We have revised all references according to the NHESSD journal style. The reference list has been updated as well. Please see references section.

The text of the manuscript has been revised.

Please also note the supplement to this comment:
http://www.nat-hazards-earth-syst-sci-discuss.net/nhess-2016-391/nhess-2016-391-AC4-supplement.pdf

[Figure]

**Supplement:**

[revised manuscript text omitted]
 \quad h_i = 0\\ \frac{\rho_i}{\rho_a} h_i \frac{U}{l} & if \quad k_i l \ge h_i\\ \frac{\rho_i}{\rho_a} k_i U & if \quad k_i l \le h_i \end{cases}$$
(2)

wherein,  $h_i$  represents the thickness of the *i* th layer of the landslide in the 118 movement process;  $\rho_i$  represents the density of the *i* th layer of the landslide in the 119 movement process;  $\rho_a$  represents the density of the landslide; the dimensionless 120 parameter  $k_i$  represents the entrainment rate.

**121 The momentum balance equation is:**

$$\partial_t \left( HU_x \right) + \partial_x \left( HU_x^2 + \frac{g_z k_{a/p} H^2}{2} \right) + \partial_y \left( HU_x U_y \right) = S_{gy} - S_f \left( R \right) \left[ n_x \right]$$
(3)

$$\partial_{t} \left( HU_{y} \right) + \partial_{y} \left( HU_{y}^{2} + \frac{g_{z} k_{a/p} H^{2}}{2} \right) + \partial_{x} \left( HU_{x} U_{y} \right) = S_{gx} - S_{f} \left( R \right) \left[ n_{y} \right]$$
(4)

[revised manuscript text omitted]

---

## Author Comment (AC7) · 28 Jun 2017

Response to Reviewer Comments

Manuscript title: (the original title: Hazard Assessment Comparison of Tazhiping Landslide Before and After Treatment) Manuscript number: 2016-391 Thanks very much for reviewer's comments, which helped us to improve the quality of manuscript. We have made major revisions to address the comments raised by the reviewer. The following responses have been prepared to address reviewer' s comments in a point-by-point fashion. All changes have been marked with RED in the revised manuscript. We would be happy to make further modifications if required. We hope the changes listed have made the manuscript suitable for publication, and we look forward to your response.

[Figure]

General comments

Q1: The paper by Huang et al. addresses relevant scientific and technical questions. It presents a concept and adoption of a well-known method to simulate mass movement processes. The used methods are in principle up to international standards but there is some doubt whether they used the appropriate method for this study.

A1: We acknowledge the remark of the reviewer. This paper addresses an interesting and relevant scientific topic. We adopted a well-known method to simulate mass movement processes. The methods are up to international standards. We are expanding the method in this contribution.

Q2: The scientific methods and assumptions used are valid and outlined clearly. There is some confusion about the mass movement process that is discussed and approached by the presented and adopted rheological model. In principle, the numerical approach in RAMMS can also be used for the simulation of landslides. But it is actually not intended for it and does not take into account specific properties of this kind of mass movement (landslides).

A2: We totally agree with the reviewer. The geomorphic heterogeneity of rapid shallow flow-type landslides such as hillslope debris flows is larger than those observed in channelized debris flows; however, many of these flows can be successfully modeled using the Voellmy-fluid friction relation with a block release initiating the flow (Christen et al., 2012). Therefore, the numerical approach in RAMMS can be used for simulation of flow-type landslides. We have added new interpretation in the new version of the manuscript. Please see Page 2, line 63-64 and the discussion section in the new version of the manuscript.

Q3: The results of the study are not really surprising. Interpretation of the simulation results is derived poorly. Some important questions remain still unanswered, namely the sensitivity of the friction parameters and more important the derivation of the best-fit parameters presented in Table 2. This aspect should be at least considered in the discussion and ideally in the methods section. While the methods section is very detailed (and also well written in good English) regarding the numeric, no information is given about the modeling procedure and interpretation of the simulation results.

A3: The result of the study is in agreement with field survey results. The derivation of the best-fit parameters is an important issue, which we have elaborated on in the discussion and methodology sections. The present estimation of model parameters can be acquired by laboratory or small-scale experiments in some instances, however calculation of the Voellmy rheological model friction coefficient is difficult. Therefore, we tested different coulomb friction coefficient values ranging between and viscous friction coefficient values ranging between . Finally, we selected the coulomb friction coefficient and viscous friction coefficient in accordance with back-analyses of well-documented landslides (Cepeda, J., et al. 2010; Du et al., 2015 ). The methods and discussion sections have been revised in the new version of the manuscript. Thank for your compliment on the writing of methods section in the new version of the manuscript. Please see p.12-13, line 271-278 and p.22-23, line 384-411.

Q4: The title does not promise detailed information about the numeric but rather a specification about the hazard assessment comparison. Therefore or the title or the content of the paper should be changed. The same is true for the abstract. More information should be given for the methods section or the method section should be adjusted. The mathematical formulae, symbols, abbreviations and units are correctly defined and used.

A4: Thank you for the insightful comments. We totally agree that the title should be revised. The title of this paper has been revised to "Hazard Assessment Comparison of Tazhiping Landslide Before and After Treatment Using the Finite Volume Method". The corresponding abstract has been revised as well. The methods section has been revised. Please see the response to comment A3. Please see p.1, line 2 and line 12-13; p.12-13, line 271-278.

Q5: There is some confusion in terminology for figures 6 and 7, that have to be changed. Figures should be improved. Figure 1 seems to be taken from an existing paper without citation. Figure 2 needs more information about the location of the study site in a global perspective and better visualization of the exact location in the Baisha river basin. figures 6 and 7 do not contain more details on the landslide area, location of the objects at risk, etc. This information is only given in figure 8 but visualized rather small. Readability of the outlines of buildings is very hard and not mentioned in the legend. The authors give in principle proper credit to previous and related work. Own contributions are not well indicated (besides the adoption of the model and the interpretation of the simulation results).

A5: Thank you for pointing out the accurate terminology. The confusion in terminology for Figures 6 and 7 has been revised. Please see p.14, line 282 and 284; p.16, line 323 and p.17, line 325. We have re-organized and added more information about the location of the study area. The Baisha river basin is visible in Figure 2. Please see p.9, line 214-215. In Figures 6 and 7 we added more detail on the landslide area shown in Figures 7a and 9a. Please see p.13, line 281; and p.16, 322. Figure 10 has been extensively revised. Building outline were added to the legend. Please see p.21, line 376 and p.22, line 379. Various minor modifications and revisions were made to all Figures. Please see p.10, line 216-217 and 218-219.

Q6: Number and quality of the references are appropriate. There are some publications in Chinese that are not accessible by all fellow scientist. There is some confusion for the article by Zhang,Z.Y., Wang,S.T., Wang,L.S.,et al., about the year of publication. In the text 1994 is mentioned while in the references there is written 1993. The reference of Toro, 1992 is missing.

A6: We have deleted some parts of unimportant Chinese's literature and revised all references according to the NHESSD journal style. We have cited the reference of Toro, 1992. Please see references section in the new version of the manuscript. Please see p.2, line 46, 51 and 52.

[Figure]

Q7: Structure and length of the paper is adequate. Methods section with the numeric is too long compared to the results section.

A7: We appreciate the comments. The methods section is very detailed; no more information is given on the simulation results. Thus, we have added more interpretation into the results section. Please see the results section in the new version of the manuscript.

Q8: Technical language and the English is more or less of good quality and understandable. Several sentences need to be reformulated, mostly because of wrong word order. There is no supplementary material available.

A8: We have carefully proofread the whole manuscript to exclude language issues as much as possible. All changes have been marked with BULE in the revised manuscript.

Other specific comments are given below.

Q9: p.2, line 61: what do the authors exactly mean with "landslide-debris flows?" Please rely on some definitions in the literature.

A9: Landslides move downslope in many different ways (Varnes, 1978). Flow-type landslides can evolve into rapidly travelling flows, which exhibit characteristics of debris flows on unchannelized or only weakly channelized hillslopes. The geomorphic heterogeneity of rapid shallow flow-type landslides such as hillslope debris flows is larger than those observed in channelized debris flows, however, many of these flows can be successfully modelled using the Voellmy-fluid friction relation and with an initial block release (Christen et al., 2012). It is true that there is some confusion about the term "landslide-debris flows" used here. We have revised it to "flow-type landslides" and add some definitions from the literature. Please see p2, line 63-64 and discussion section p22 and 23.

Q10: p.2, line 71: what to the authors exactly mean with 3D mapping of the division of hazard zones? Usually, hazards zonation is given on a map, e.g. in 2D

A10: Thank you for the correction. It has been revised to 2D. Please see p.2, line 74.

Q11: p.3, line 98: this figure is taken from Christen et al., 2010. Please cite source.

A11:It has been added. Please see p.3, line 101.

Q12: p.3, line 107: missing space.

A12: It has been revised. Please see p.4, line 109.

Q13: p.7, line 178: this reference is missing in the reference section.

A13: It has been cited. Please see reference p.26, line 539-540 and p.7, line 178 -179. " Toro, E.F.: Riemann problems and the waf method for solving the two dimensional shallow water equations, Philos. Trans. R. Soc. London, Ser., A 338, 43–68. 1992".

Q14: p.11, line 255: see comment for p.2, line 71

A14: It has been revised to 2D. Please see p.12, line 263.

Q15: p.11, line 266: figure is subtitled with "Thickness". Thickness of deposition is not equal to flow height (if a landslide really "flows"...). Please adapt wording.

A15: Thank you for pointing out the inaccurate terminology. It has been revised to flow height. Please see p.1, line 15; see p.14, line 282; p15, line 292 and 294; p.16, line 323; p17, line 332 and 334; p18, line341; p23, line 416.

Q16: p.12, line 268: subtitle of figure is "Speed", legend says "Velocity". If the blue to green marked zone shows the deposited mass of the landslide, there should be no velocity value (because it's deposited). In chapter 3 is no indication or estimation about the speed of the landslide mass, therefore figure 6b does not really make sense.

A16: Thank you for pointing out the inaccurate terminology. It has been revised to Velocity. Please see p.1, line 15; p.14, line 284; p15, line 295 and p.17, line 325 and 334. In any case, velocity plays a more important role regarding kinetic energy acting on an obstacle. However, the Miaoba residential area of Red Village is located at the frontal part of Tazhiping landslide. Therefore, the maximum flow height of the landslide

is one of the direct factors influencing the building's deformation failure status. Please see p.18, line 346-353.

Q17: p.12, line 270: not clear, if the colored area shows the maximum pressure or an instantaneous for a given time step. Much more of interest would be a local value (over time) at the position of a building. And why the legend goes up to more than 1000kPa but no reddih or yellowish areas are marked?

A17: The colored area shows the maximum values of moving process or an instantaneous pressure for a given time step. As the building of Red Village is located at the frontal part of landslide, the pressure of the middle and lower landslide deposits was about 200kPa. Thus, three-story and lower houses within the deposition range might be buried. The maximum pressure value in the surface gully can be found in the middle and upper slope. According to field surveys, we have found this gully is at an elevation of about 1,200 m. The maximum pressure value is easily found from the instantaneous for a given time step figure. Therefore, coupled with field observations and numerical simulation, they are especially helpful in understanding the landslide movement process in complex terrain. It has been introduced in p.17, line 329-330.

Q18: p.12, lines 274, 277 and p.13, line 278: not clear what numbers in the circle mean. Is this kind of a list or does it indicate a location in a figure?

A18: No, it does not indicate a location. It has been deleted. Please see p.14, line 291; p.15, line 294 and 295.

Q19: p.13, line 279: how is made this separation between houses of different numbers of stories? Please give more information and references to it.

A19: The building is 3m high on each floor. We have cited some literature (Hungr et al., 1984; Petrazzuoli et al., 2004; GB, 50010–2010; Hu et al., 2012; Zeng et al., 2015). Please see p18, line 362 and 364.

Q20: p.13, line 293: or indicate "about 1.2 m" or give exact value.

A20: The more exact value has been given. " with an elevation of 1,070-1,072m and a length of 182m." Please see p.15, line 317-318.

Q21: p.13, line 298: same remark as for figure 6a.

A21: It has been revised. Please see A.15.

Q22: p. 14, line 300: same remark as for figure 6b

A22: It has been revised. Please see A.16.

Q23: p.14, line 305: example of a sentence that has to be rewritten because of wrong word order.

A23: We have revised the sentence to read "Provided in Fig.9 are the kinematic characteristics of the landslide deposit." Please see p.17, line 329.

Q24: p.14, lines 305, 308, 309: not clear what numbers in the circle mean.

A24: It has been deleted. Please see p.17, line 331 and 334; p.18, line 335.

Q25: p.15, line 321/322: not sure, if this statement is really true. There may be examples where entire houses on a landslide mass are moved but not destroyed because of stable base plates. In any case, velocity plays a more important role regarding kinetic energy acting on an obstacle. You are right in the sense that the height of a moving landslide (e.g. the frontal part) plays an important role when it hits a building on a higher level, e.g. the second or third floor. Please clarify this point.

A25: We have clarified this point. "Landslides reflect landscape instability that evolves over meteorological and geological timescales, and they also pose threats to people, property, and the environment. The severity of these threats depends largely on landslide speed and travel distance. There may be examples where entire houses on a landslide mass are moved but not destroyed because of stable base plates. In any case, velocity plays a more important role regarding kinetic energy acting on an obstacle. However, the Miaoba residential area of Red Village is located at the frontal part

of Tazhiping landslide." Please see p.18, lines 346-353.

Q26: p.15, 16 and 17, table 3: the term "washed away" is not suitable for landslide process. It implies an major influence by a fluid.

A26: It has been revised. Please see p.18, line 366.

Q27: p.17, line 333: This should be 2D, because you show a map with the different zonations. These different zonations are not defined, by the way.

A27: It has been revised to 2D. Please see p.20, line 367 and p.22, line 381.

Q28: p.17, line 339: There seem to be marked buildings (in the red high-hazard zone). If so, adjust legend and make sure they are better visible. What zone is defined outside the colored area? No hazard or also low-hazard zone?

A28: In figure 10, the red high-hazard zone of buildings has been marded. We have adjusted the legend and defined outside the colored area as no-hazard. Please see p.20, line 371-374 and Figure.10 legend.

Q29: p.18, line 342: same as for figure 8a. And this should be 8b instead of 8c.

A29: It has been revised. Please see p.22, line 381.

Q30: p.18, line 350: what is a landslide-debris flow?

A30: It has been defined. Please see p.23, line 389-396 and answer A5.

Q31: p.18, line 358: this should be 2D.

A31: It has been revised. Please see p.23, line 423.

Q32: p.19, line 411: correct reference would be: Michael-Leiba, M., Baynes, F" Scott, G., Granger, K. 2003. Regional landslide risk to the Cairns community [J]. NatHazards, 2003,30 (2):233–249. Check reference style for all references according to the journal style!

A32: We have revised all references according to the NHESSD journal style. The reference list has been updated as well. Please see references section.

The text of the manuscript has been revised.

Please also note the supplement to this comment:
https://www.nat-hazards-earth-syst-sci-discuss.net/nhess-2016-391/nhess-2016-391-AC7-supplement.pdf
* * *
[Figure]

**Supplement:**

**Hazard Assessment Comparison of Tazhiping Landslide Before and After Treatment using the finite volume method**

Dong Huang [1], YuanJun Jiang[1]*,JianPing Qiao[1], Meng Wang[1]

1. Key Laboratory of Mountain hazards and Surface process, Institute of Mountain hazards and Environment, Chinese Academy of Science, Chengdu 610041, China
*Corresponding author ( yuanjun.jiang.civil@gmail.com).

**Abstract:** Through investigation and analysis of geological conditions and mechanical parameters of the Taziping landslide, the finite volume method was adopted, and, the rheological model was adopted to simulate the landslide and avalanche entire mass movement process. The present paper adopted the numerical approach of RAMMS and the GIS platform to simulate the mass movement process before and after treatment. This paper also provided the conditions and characteristic parameters of soil deposits ( flow height,  velocity, and stresses) during the landslide mass movement process and mapped the 3D division of hazard zones before and after landslide treatment. Results indicated that the scope of hazard zones contracted after engineering treatment of the landslide. The extent of high-hazard zones was reduced by about 2/3 of the area before treatment, and characteristic parameters of the mass movement process after treatment decreased to 1/3 of those before treatment. Despite engineering treatment, the Taziping landslide still poses significant hazard to nearby settlements. Therefore, we propose that houses located in high-hazard zones be relocated or reinforced for protection.

**Keywords**: finite volume method; rheological model; motion feature parameters; hazard assessment

**1. Introduction**

The hazards of a landslide include scope of influence (i.e., source area, possible path area, and backward and lateral expansion area) and secondary disasters (i.e., reservoir surge, blast, and landslide-induced barrier lake). A typical landslide hazard assessment aims to propose a systematic hazard assessment method with regard to a given position or a potential landslide. Current research on typical landslide hazard assessment remains immature, and there are multiple methods for interpreting landslide hazards. To be specific, the scope of influence prediction of a landslide refers to deformation and instability characteristics such as sliding distance, movement speed, and bulking thickness range. The movement behavior of a landslide mass is related to its occurrence, sliding mechanisms, mass characteristics, sliding path, and many other factors. Current landslide movement prediction methods include empirical prediction and numerical simulation.

**Empirical prediction method:** The empirical prediction method involves

[a1]: Answer to the comment Q4: The title of this paper has been revised and added information about the numeric.

[a2]: Answer to the comment Q4: The abstract has been revised and added information about the numeric.

[a3]: Answer to the comment Q15: It has been revised to flow height and velocity.

analyzing landslide flow through the collection of landslide parameters in the field. It further consists of the geomorphologic method (Costa, 1984; Jackson et al., 1987; Scott et al., 1993), the geometric change method ( Finlay et al., 1999; Michael-Leiba et al., 2003), and the volume change method (Fannin et al., 2001). Empirical models are commonly simple and easy to apply, and the required data are easy to obtain as well. **Numerical simulation method:** Numerical simulation methods are further divided into the continuous deformation analysis method (Hungr, 1995; Evans et al., 2009; ; Wang , et al., 2016), the discontinuous deformation analysis method (Shi , 1988; ), and the simplified analytical simulation method (Christen et al., 2010a; Sassa, 2010; Bartelt et al., 2012; Du et al., 2015). The numerical simulation method expresses continuous physical variables using the original spatial and temporal coordinates with geometric values of discrete points. Numerical simulations follow certain rules to establish an algebraic equation set in order to obtain approximate solutions for physical variables.

Empirical prediction models only provide a simple prediction of the sliding path. Due to the differences in geological environments, empirical prediction models commonly have low generality. The continuous deformation method has the advantage of an extremely strong replication capability, but it is not recommended when analyzing flow-type landslides , lahars, or debris flows because of complicated rheological behaviors (Iverson et al., 1997, 2001; Hungr et al., 2001; Glade 2005; Portilla et al., 2010; Chen et al., 2014). The fluid mechanics-based discontinuous deformation method has several shortcomings such as, great computational burden, difficult parameter selection, and difficult 3D implementation. The simplified analytical simulation method fully takes into account the flow state properties of landslides before introducing a rheological model and can easily realize 3D implementation on the GIS platform. On that account, this paper adopted the continuous fluid mechanics-based finite volume method (simplified analytical simulation method). We introduce a rheological model on the basis of using mass as well as momentum and energy conservation to describe the movement of landslides. We also employed GIS analysis to simulate the entire movement process of Taziping landslide and map the 2D division of hazard zones.

**2. Methods**

**2.1 Kinetic analysis method**

Adopting the continuous fluid mechanics-based finite volume method, this paper took into account erosion action on the lower surface of the sliding mass and the change in frictional resistance within the landslide-debris flow in order to establish a computational model. The basic idea is to divide the calculation area into a series of non-repetitive control volumes, ensuring that there is a control volume around each grid point. Each control volume is then integrated by the unresolved differential equation in order to obtain a set of discrete equations. The unknown variable is the numerical value of the dependent variable at each grid point. To solve the integral of a control volume, we make a hypothesis about the change rule of values among grid

**[a4]:** Answer to the comment Q6: We have deleted some parts of unimportant Chinese's literature and revised all references according to the NHESSD journal style.

**[a5]:** Answer to the comment Q6 and Q32: We have deleted some parts of unimportant Chinese's literature and revised all references according to the NHESSD journal style.

**[a6]:** Answer to the comment Q6 and Q32: We have deleted some parts of unimportant Chinese's literature and revised all references according to the NHESSD journal style.

**[a7]:** Answer to the comment Q9: It is true that there is some confusion about the term "landslide-debris flows" we used here. We have revised it to "flow-type landslides".

**[a8]:** Answer to the comment Q2 and Q9: We have added some definitions in the literature. The more detailed definition in the discussion part.

**[a9]:** Answer to the comment Q10: It has been revised to 2D.

[revised manuscript text omitted]

[a28]: Answer to the comment Q15: we have revised the terminology.

[a29]: Answer to the comment Q8: We have carefully proofread the whole manuscript to exclude language issues as much as possible.

[a30]: Answer to the comment Q15: we have revised the terminology.

[a31]: Answer to the comment Q18: The numbers in the circle does not indicate a location. It has been deleted.

[a32]: Answer to the comment Q16: It has been revised to

[a33]: Answer to the comment Q18: The numbers in the circle does not indicate a location. It has been deleted.

[a34]: Answer to the comment Q19: The building is 3m height each floor.

[a35]: Answer to the comment Q20: The more exact value has been given.

[Figure]

                                             (a)  Flow height

[a36]: Answer to the comment Q5 and Q21: we added more details on the landslide area, and revised the terminology.

[Figure]

(b)  Velocity

[Figure]

(c) Pressure

Fig.  9 Movement characteristic parameters of the Taziping landslide (after treatment)

Provided in Fig.  9 are the kinematic characteristics of the landslide deposit. The colored bar shows the maximum values of the kinematic process for a given time step.

① Deposits accumulated during the landslide movement process had a maximum flow height of 18.37m, located around the surface gully of the middle and upper slope. The middle and lower portions of the landslide deposit had a flow height of approximately 3-5m. ② The middle and lower movement

[a37]: Answer to the comment Q22: It has been revised to

[a38]: We have added Figure4 and Figure8 .

[a39]: Answer to the comment Q23: The sentence be rewritten because of wrong word order.

[a40]: Answer to the comment Q17: The colored bar shows the maximum values of moving process or an instantaneous for a given time step. According to field surveys, we have found this gully is at an elevation of about 1,200 m. The maximum pressure value is easily found from the instantaneous for a given time step figure. Therefore, coupled with field observations and numerical simulation, they are especially helpful in understanding landslide movement process in complex terrain.

[a41]: Answer to the comment Q24: The numbers in the circle does not indicate a location. It has been deleted.

[a42]: Answer to the comment Q21: we have revised the terminology.

[a43]: Answer to the comment Q8: We have carefully proofread the whole manuscript to exclude language issues as much as possible.

[a44]: Answer to the comment Q21: we have revised the terminology.

[a45]: Answer to the comment Q24: The numbers in the circle does not indicate a location. It has been deleted.

[revised manuscript text omitted]

[a55]: We have added Figure4 and Figure8 .

[a56]: Answer to the comment Q27: It has been revised to 2D.

[a57]: Answer to the comment Q28: We have defined outside the colored area as no-hazard.:

[Figure]

[Figure]

                          **(a) Before treatment**

**[a58]:** Answer to the comment Q5 and Q28: This figure has been extensively visualized and added the outlines of buildings in the legend. We have adjusted legend and marked the red high-hazard zone of buildings.

[Figure]

[Figure]

**(b) After treatment**

**Fig.10 2D division comparison of the hazards of the Taziping landslide**

**5 Conclusions and Discussion**

The hazard assessment of landslides using numerical models is becoming more
and more popular as new models are developed and become available for both

[a59]: Answer to the comment Q5 and Q28: This figure has been extensively visualized and added the outlines of buildings in the legend. We have adjusted legend and marked the red high-hazard zone of buildings.

[a60]: Answer to the comment Q29: p.18, line 342: same as for figure 8a. And this should be 8b instead of 8c.It has been revised.

[a61]: We have added Figure4 and Figure8 .

[a62]: Answer to the comment Q27: It has been revised to 2D.

scientific research and practical applications. There is some confusion about the mass movement process that is discussed by the rheological model presented in this contribution.

Landslides move downslope in many different ways (Varnes, 1978). In addition, landslides can evolve into rapidly travelling flows, which exhibit characteristics of debris flows on unchannelized or only weakly channelized hillslopes. The geomorphic heterogeneity of rapid shallow landslides, such as hillslope debris flows, is larger than observed in channelized debris flows; however many of these flows can be successfully modelled using the Voellmy-fluid friction (Christen et al., 2012). Results presented in this paper support the conclusion that Voellmy-fluid rheological model can be used to simulate flow-type landslides.

The selection of model parameters remains one of the fundamental challenges for numerical calculations of natural hazards. At present, there are numerous empirical parameters obtained from 30-years of monitoring data. Such as in RAMMS, we can automatically generate the friction coefficient of an avalanche for our calculation domain based on topographic data analysis, forest information and global parameters (WSL, 2013). The friction parameters for debris flows can found in some literature (Fannin et al., 2001; Iovine et al., 2003; Hürlimann et al., 2008; Scheidl et al., 2010; Huang et al., 2015). However, there is little research regarding friction parameters of flow-type landslide. Therefore, we tested different coulomb friction coefficient $\mu$ values ranging between $0.1 \leq \mu \leq 0.6$ and viscous friction coefficient $\zeta$ values ranging between $100 \leq \mu \leq 1000 m \cdot s^{-2}$. Finally, we selected the coulomb friction coefficient $\mu = 0.45$ and viscous friction coefficient $\zeta = 500 m \cdot s^{-2}$ in accordance with back-analyses of well-documented landslides (Cepeda et al., 2010; Du et al., 2015). Simulation results are consistent with field observations of topography and sliding path.

Based on the finite volume method and the RAMMS program, the simulation results of Taziping landslide were consistent with the sliding path predicted by the field investigation. This correlation indicates that numerical simulation is an effective method for studying the movement processes of flow-type landslides debris flows. The accumulation thickness flow height and pressure of landslide deposits were reduced by about 1/2, and the kinematic speed was reduced by about 1/3 after treatment. However, the Miaoba residential area of Red Village is still partially at hazard. Considering that two-story and lower houses within the deposition range might be buried, it was further suggested that the design strength of the gable walls of houses on the middle and upper parts of the deposit be increased above 150kPa.

By utilizing a GIS platform in combination with landslide hazard assessment indexes, we mapped the 32D division of the Taziping landslide hazard zones before and after engineering treatment. The results indicated that overall hazard zones contracted after engineering treatment and, the area of high-hazard zones was reduced by about 2/3. After engineering treatment, the number of at hazard houses on the left

[a63]: Answer to the comment Q3 , Q9 and Q30: The text in the discussion section have been revised. what is a landslide-debris flow? It has been defined.

[a64]: Answer to the comment Q9: It is true that there is some confusion about the term "landslide-debris flows" we used here. We have revised it to "flow-type landslides".

[a65]: Answer to the comment Q15: we have revised the terminology.

[a66]: Answer to the comment Q31: It has been revised to 2D.

[revised manuscript text omitted]

Zeng, C., Cui, P., Su, Z.M., Lei, Y., Chen, R.: Failure modes of reinforced concrete columns of buildings under debris flow impact, Landslides., 12, 561-571, 2015.

[a68]: Answer to the comment Q32: We have revised all references according to the NHESSD journal style. The reference list has been updated as well.

[a69]: Answer to the comment Q13: It has been cited.

[a70]: Answer to the comment Q6: We have deleted some parts of unimportant Chinese's literature and revised all references according to the NHESSD journal style.

---

## Author Comment (AC8) · 28 Jun 2017

Response to Reviewer Comments

Manuscript title: (the original title: Hazard Assessment Comparison of Tazhiping Landslide Before and After Treatment) Manuscript number: 2019-391 Thanks very much for reviewer's comments, which helped us to improve the quality of manuscript. We have made major revisions to address the comments raised by the reviewer. The following responses have been prepared to address reviewer' s comments in a point-by-point fashion. All changes have been marked with RED in the revised manuscript. We would be happy to make further modifications if required. We hope the changes listed have made the manuscript suitable for publication, and we look forward to your response.

[Figure]

General comments

Q1: The manuscript presented a fluid mechanics based method for landslide/debris flow modeling, and was further applied to a real landslide case for hazard zones mapping. The topic is scientifically significant for nature hazard mitigations. The manuscript was logically organized and the results were well described and reasonably discussed. The authors provided sufficient evidence that the proposed method could be used as a promising tool for landslide modeling and hazard mapping. The knowledge obtained from the study would benefit civil engineering society for landslide investigation assessment. This paper can be accepted for publication by considering all the points given below.

A1: Thank you very much for the reviewer's positive comments about our work. We have addressed each comment meticulously and illuminated the requests in the following responses and the text as much as possible.

Q2: The main contribution of this paper seems to be the computational model proposed. It is desired to add related descriptions to the title of this paper.

A2: We appreciate the reviewer's suggestion. The title of this paper has been revised to "Hazard Assessment Comparison of Tazhiping Landslide Before and After Treatment Using the Finite Volume Method". Please see p.1, line 2.

Q3: Previous study on landslide/debris flow issues using the fluid mechanics based method had faced the problem that it predicts higher mobility of the moving body while using the same fluid parameters throughout the whole flowing process. For example, less obvious fluid property is expected when the flow body is approaching stop point. It is stated in this manuscript that a changed frictional resistance is used (L78). However, the details are not clear in the text. Relevant descriptions on this issue should be strengthened.

A3: This paper adopted the RAMMS to simulate the mass movement process. In

RAMMS, the friction coefficient for our calculation domain can be automatically adjusted based on topographic data analysis, forest information and global parameters. Therefore, a changed frictional resistance was applied to the slide mass during the flowing process. We added more details in the discussion section. Please see p.22-23, line 384∼411.

Q4: It is not clear in the text that how the free surface of the landslide/debris flow is treated or reconstructed. An additional figure is need to describe the details.

A4: The landslide body as well as the calculation domain were reconstructed and specified though the topographic data input with the built-in RAMMS Project Wizard. We have added new Figure 4 to show more details. Please see p.10, line 219.

Q5: Fig.4 showed the geological profile of Taziping Landslide and a slide surface is clearly indicated. Is this slide surface comparable with the simulation result? It would be interesting to show their comparison.

A5: The indicated slide surface in Fig.4 shows a potential surface before treatment. Combined with the other field survey data. It was concluded that the sliding mass had an estimated starting volume of about 600,000m3 and a mean thickness of 8m. After fully accounting for the slide-resistant piles and mounds, we introduced the Morgenstern-Price method to calculate the stability coefficient of Taziping landslide after treatment. The method was determined with an iterative approaching by changing the position of the sliding surface until failure of the dumpsite (Figure. 8). It was suggested the treatment significantly improved the slide stability. We added more descriptions on this issue. Please see p.15, line 305∼307 and 311∼312.

Q6: In Tab.3, Various hazard zone levels were cataloged. What is the criterial to assign a specific damage situation to a certain zone level? Is there any standard code to follow?

A6: The hazard zone levels were cataloged according to current standard and literatures. We have cited the relevant standard code and literature (Fell R et al., 2008; DZ/T 0286-2015). Please see p.18, line 359~360.

Other specific comments are given below.

Q7: The quotations in the manuscript are not in the same format, for example, Line 44, Costa, 1984; VS Line 50, Zhang. Y, 2013. Usually only family name is preferred, please refer to the journal's instructions and make necessary changes throughout the text.

A7: Thank you for pointing out the inaccurate quotation. It has been revised. We have revised all references and quotations in the manuscript according to the NHESSD journal style. The reference list has been updated as well. Please see the references section.

Q8: Fig.1 needs proper citation.

A8: Thank you for the correction. It has been revised (Christen et al., 2010a).

Q9: In Fig.6, Fig.7, what moment of flow does these figures represent? Different moment should have different deposit thickness, flow velocity and pressure. Please confirm.

A9: Figure 6 and Figure 7 show the last moment of the flow. The flow has a different deposit flow height, velocity and pressure at various moments in time. However, the colored bar shows the maximum values of the movement process or an instantaneous for a given time step. It has been revised. Please see p.17, line 329-331.

Q10: L276 "The middle and lower deposits had a thickness of about 5-10m", confusing here, what does "the middle and lower deposits" mean? Similar as "the middle and lower movement speed", please check throughout the text.

A10: The authors apologize for the confusion. The sentences have been reformulated. Please see p.15, line 293 and 294; p.17, line 333.

Q11: L289. What technique is used for searching the sliding plane.

A11: The method coupled with field borehole surveying and the numerical calculation method described in Q5 were used to search the sliding plane.

Q12: L305, Fig.4 should be Fig.7.

A12: Thank you for the correction. It has been revised. Please see p.17, line 328.

Q13: Tab.3. How is the "Building damage probability" evaluated?

A13: Thank you for the comment. Building damage probability is evaluated by the thickness of a landslide mass that the building can withstand. We have cited the relevant literature (Hungr et al., 1984; Petrazzuoli et al., 2004; GB, 50010–2010; Hu et al., 2012; Zeng et al., 2015). Please see p18, line 362 and 364.

The text of the manuscript has been revised.

Please also note the supplement to this comment:
https://www.nat-hazards-earth-syst-sci-discuss.net/nhess-2016-391/nhess-2016-391-AC8-supplement.pdf

**Supplement:**

[revised manuscript text omitted]

[a8]: Answer to the comment Q10: This sentences has been reformulated, because of wrong word order.

[a9]: Answer to the comment Q5 and Q11: Before engineering treatment, Figure.4 and Figure.5 have showed that the sliding mass had an estimated starting volume of about 600,000m$^3$ and a mean thickness of 8m. After fully accounting for the slide-resistant piles and mounds, we introduced the Morgenstern-Price method to calculate the stability coefficient of Taziping landslide after treatment. The method was determined with an iterative approaching by changing the position of the sliding surface until failure of the dumpsite (Figure.8)

[a10]: Answer to the comment Q5: We have reconstructed and added Figure8.

[a11]: Answer to the comment Q5:The result of numerical analysis.

[Figure]

                 (a)  Flow height

(b)  Velocity

[Figure]

(c) Pressure

Fig. 9 Movement characteristic parameters of the Taziping landslide (after treatment)

Provided in Fig. 9 are the kinematic characteristics of the landslide deposit. The coloredbar shows the maximum values of moving process or an instantaneous for a given time step.① Deposits accumulated during the landslide movement process had a maximum  flow height of 18.37m, located around the surface gully of the middle and upper slope. The middle and lower portions of the landslide deposit had a  flow height of approximately 3-5m. ② The middle and lower movement

**[a12]:** Answer to the comment Q12: It has been revised.

**[a13]:** Answer to the comment Q9: Figure.9 is shown that the last moment of the flow. Different moment have different deposit flow height, velocity and pressure. However, the coloredbar shows the maximum values of mowing process or an instantaneous for a given time step. It has been revised.

**[a14]:** Answer to the comment Q10: This sentences has been reformulated, because of wrong word order.

velocity of the landslide deposits ranged between 3m/s and 5m/s. ③ The
landslide had a mean pressure of about 330kPa, and the pressure of the middle and
lower deposits was about 100kPa. Thus, it could be held that two-story and lower
houses within the deposition range might be buried. It  is further suggested that
the design strength of the gable walls of houses on the middle and upper parts of the
deposits be increased above 150kPa.

After treatment, the accumulation  flow height and pressure of the
deposits were reduced by about 1/2, and the kinematic speed was reduced by about
1/3. However, the Miaoba residential area of Red Village was still partially at hazard.

**4 Results**

Landslides reflect landscape instability that evolves over meteorological and
geological timescales, and they also pose threats to people, property, and the
environment. The severity of these threats depends largely on landslide speed and
travel distance. There may be examples where entire houses on a landslide mass are
moved but not destroyed because of stable base plates. In any case, velocity plays a
more important role regarding kinetic energy acting on an obstacle. However, the
Miaoba residential area of Red Village is located at the frontal part of Tazhiping
lanslide. During landslide movement, the spatial scale indexes of a landslide mass
include area, volume, and thickness. The maximum thickness of the landslide is one
of the direct factors influencing the building's deformation failure status. A large
landslide displacement may lead to burial, collapse, or deformation failure of the
building, and thus influence its safety and stability. Thus, landslide thickness
constitutes an important index for assessing the hazards of a landslide disaster, and for
influencing the consequences faced by disaster-affected bodies (Fell et al., 2008;
DZ/T, 0286-2015). Provided in Tab.3 is a landslide thickness-based division of the
predicted hazard zones of Taziping landslide, in which the thickness of the landslide
mass correlates with the ability of a building to withstand a landslide disaster (Hungr
et al., 1984; Petrazzuoli et al., 2004; Glade 2006; GB, 50010–2010; Hu et al., 2012;
Zeng et al., 2015). After treatment with slide-resistant piles, the hazard of a future
slide was reduced by about 1/3 overall and by 2/3 in high-hazard zones.

**Tab.3 Division table of the predicted hazards of Taziping landslide (unit: m$^2$)**

| Hazard zone level | Assessment index | Building damage probability | Area before treatment | Area after treatment | Increased/decreased area | Building damage characteristics |
|---|---|---|---|---|---|---|
| **Low-hazard zone (l)** | $h \leq 0.5$m | 20% | 44 , 600 | 38 , 748 | -5,852 | One-story houses may be damaged; houses on the |

**[a15]:** Answer to the comment Q6: We have cited standard code and literature.

**[a16]:** Answer to the comment Q13:By the thickness of the landslide mass to evaluate the ability of a building to withstand a landslide disaster. We have cited relevant literatures.

[revised manuscript text omitted]

---

## Author Response (AR3)

Response to Reviewer Comments

Manuscript title: (the title: Hazard Assessment Comparison of Tazhiping Landslide Before and After Treatment Using the Finite Volume Method) Manuscript number: 2016-391 Thanks very much for reviewer's comments, which helped us to improve the quality of manuscript. We have made major revisions to address the comments raised by the reviewer. The following responses have been prepared to address reviewer' s comments in a point-by-point fashion. All changes have been marked with BULE in the revised manuscript. We would be happy to make further modifications if required. We hope the changes listed have made the manuscript suitable for publication, and we

look forward to your response.

General comments

Q1: You followed the indications given by both reviewers and you corrected and improved your manuscript accordingly. Most of the questions are answered and changes and completions are made. But the paper is still of fair quality only. Some information is given in the wrong chapters. You talk about "confusion" in the chapter "Conclusions and discussion" but your revision does not really solve the problem. Please introduce these points in the "Introduction"! .

A1: We acknowledge the remark of the reviewer. We tried our best to improve the manuscript. We earnestly appreciate the editors/reviewers' work and hope that the corrections meet their approval. We totally agree with the reviewer. We have move p. 20 lines 373 - 380 to the section "Empirical prediction method". This would help to avoid the confusion about the mass movement process that is discussed in the Methods section. Please see p.2 line 64-70.

Q2: The content of the manuscript is now more or less OK. All necessary information is given. Figures could still be improved. The comparison between pre and post treatment results could be solved easily introducing some additional lines of main results from the pre simulations in the figures of the post simulations (e.g. with a dotted dark line or similar).

A2: We appreciate the comments. We have carefully revise the whole manuscript to follow strictly the reviewers comments as much as possible. Figure.7a has been extensively revised. Please see p.13, line 289. We have re-organized and added more information about the comparison between pre and post treatment results. Please see p.16, line 330 (Figure.9a ) and see p.22, line 388 (Figure.10b ).

Q3: Abstract: 3 times "adopted", poor linguistic quality, please reformulate and improve.

[Figure]

A3: Thank you for the insightful comments. We have re-organized and improved the abstract of the manuscript. Please see the abstract section in the new version of the manuscript.

Q4: Introduction: still not enough information to avoid the confusion that the authors mention in the Conclusions and discussions. Please introduce these points mentioned in the chapter "Conclusions and discussion" here. p.2, line 58: why not move this information to the section "Empirical prediction method"?

A4: Thank you for pointing out the problem. I have move this information to the section "Empirical prediction method". Please see p.2 line 64-70 and A.1.

Q5: Methods: I somewhat miss information about the method used to do the hazard prediction and the evaluation of the dynamic interaction with buildings. Shouldn't that be introduced here?

A5: We have introduced the method used to do the hazard prediction and the evaluation of the dynamic interaction with buildings. Please see p.7 line 198-204 in the new version of the manuscript.

Q6: Study area and data: Fig. 4: What defines the blue outline of the landslide area. Is this perimeter based on field survey?

A6: Yes, it is based on the survey report and field investigation. Please see Figure.4.

Q7: Results:I miss a proper comparison between model result and reality, e.g. outline of the landslide indicated in fig. 7 . A comparison between the two situations would be interesting. p13, line 283: delete "speed" and check blanks between words in the line above

A7: We have added more information about the comparison between model result and reality. Please see Figure.9a and Figure.10b. Thank you for the correction. It has been deleted and deleted the blanks between words in the line above. Please see p.15, line 301 and 302.

[Figure]

Q8: Conclusions and discussion: I propose to move lines 373 - 380 to the Introduction (and repeat it partially in the conclusions). This would help to avoid the confusion about the mass movement process that is discussed in the Methods section. Also, the section about the selection of model parameters has to be introduced much earlyer. This is the motivation for this paper and has to be mentioned in the Introduction.

A8: Thank you for the insightful comments. We have move p. 20 lines 373 - 380 to the section "Empirical prediction method" and simply to introduce about the selection of model parameters in the Introduction. Please see p.2 line 64-71.

The text of the manuscript has been revised.

[revised manuscript text omitted]

**2. Methods**

**[a2]:** Answer to the comment Q1and Q4: We have move p. 20 lines 373 - 380 to the section "Empirical prediction method". This would help to avoid the confusion about the mass movement process that is discussed in the Methods section.

**[a3]:** Answer to the comment Q8: We have simply to introduce about the selection of model parameters in the Introduction section.

[revised manuscript text omitted]

**[a4]:** Answer to the comment Q5: We have introduced the method used to do the hazard prediction and the evaluation of the dynamic interaction with buildings.

**3. Study area and data**

**3.1 Taziping landslide**

[revised manuscript text omitted]

[a9]: Answer to the comment Q2and Q7: We have re-organized and added more information about the comparison between pre and post treatment results could be solved easily introducing some additional lines of main results from the pre simulations in the figures of the post simulations.

[a10]: Answer to the comment Q8: We have move p. 20 lines 373 - 380 to the section "Empirical prediction method" and repeat it partially in the conclusions.

[revised manuscript text omitted]